# Fertility plans in the early times of the COVID-19 pandemic: The role of occupational and financial uncertainty in Italy

**Francesca Luppi** [1]*, **Bruno Arpino**[2], **Alessandro Rosina**[1]

**1** Department of Statistics, Università Cattolica del Sacro Cuore, Milan, Italy, **2** Department of Statistics, Computer Science, Applications, University of Florence, Florence, Italy

* francesca.luppi1@unicatt.it

**Data Availability Statement:** All relevant data are within the article and its Supporting Information files.

**Funding:** The author(s) received no specific funding for this work.

## Abstract

This study investigates the association between objective and subjective indicators of economic uncertainty, generated by the COVID-19 health and economic crisis, and young Italians' fertility plans during the 2020. We use unique repeated cross-sectional data, collected at different time points during the pandemic (March and October/November 2020) together with pre-COVID data (2016). The data offer a standard fertility intention question pre- and during-COVID, and also a direct question on whether pre-COVID fertility plans have been confirmed, postponed or abandoned. In March 2020, individuals with more vulnerable occupations show a lower probability of intending to have a(nother) child in the short-term and a higher probability of abandoning their pre-COVID fertility plan; in October 2020 changes in fertility plans do not vary by employment condition. Instead, both in March and October, those who suffered from a negative income shock and those with negative expectations on their future income and occupation are more likely to abandon their pre-pandemic fertility plan compared to their better off counterparts. Overall, economic uncertainty seems to have similarly affected men and women's fertility intentions. Our findings point to the fact that the unequal economic consequences of the pandemic also produced and will produce heterogeneous effects on fertility intentions.

## Introduction

The COVID-19 pandemic differently impacted on birth rates among high-income countries. Despite the original expectations about the fact that economic and social changes derived by the COVID pandemic would have been affecting negatively on fertility both in the short and in the long term [1, 2], early evidence is quite mixed [3, 4]. In particular, at least between December 2020 and early 2021, the COVID crisis results in a significant birth decline such as in Italy, Spain, Belgium, and Portugal [3], while an increase seems to appear in some Nordic countries [e.g., 4, 5].

Previous studies report evidence of strong drops in births during and after great epidemics, such as the 1918–19 Spanish flu worldwide [6–8] and the Zika epidemic in South America [9].

**Competing interests:** The authors have declared that no competing interests exist

Despite the 1918 and the 2020 pandemics share some similar patterns, differently from the Spanish flu, which strongly hit the young population, the COVID-19 pandemic has increased mortality especially among older people [10]. This means that the health dimension of the COVID-19 crisis may not be directly responsible for a revision of pre-COVID fertility intentions, while the subsequent economic recession may produce such an effect. Further, in periods of great economic uncertainty, as the COVID-19 pandemic has been proved to be [11], people tend to postpone childbearing [12–14], and long-lasting recessions negatively affect fertility intentions [15–18], if they lead to an enduring or even permanent loss of income among young adults [19].

In this paper we examine the role of the increased objective and subjective uncertainty link to the individual's occupational condition and income due to the COVID-19 crisis on young women's and men's fertility plans in Italy during the year 2020. Consistently with our previous considerations, we expect a higher probability of withdrawn or postponed fertility plans among young individuals with more precarious income and employment conditions, while the incidence of the COVID epidemic is expected to play, in case, a more marginal role.

Although some official statistics on births are available at the aggregate level (see e.g., [20] for Italy, [21, 22] for other countries), our focus on fertility intentions allows assessing the role of the perceived and experienced uncertainty about economic and occupational conditions created by the pandemic on the possible redefinition of individuals' fertility plans at the micro level. Macro-level fertility trends are informative on the overall possible short-term fertility effects of the pandemic, but they are limited in their ability to provide insights on the heterogeneities in these effects, which is crucial given that the pandemic's effects have been considerably unequal (e.g., [22–26]). Our micro-level data, instead, allow a timely overview about how income and occupational uncertainty has affected fertility plans in the early phase of the Covid crisis, highlighting which part of the young population might be targeted of ad hoc policies to contrast a further decline in births during the next years.

We use a unique longitudinal (repeated cross-sectional) dataset, collected at two different time points during the pandemic (March and October/November 2020) together with pre-COVID data (2016), to explore the changes in Italian young people's fertility intentions. Compared to panel surveys, repeated cross-sectional surveys have the advantage of providing data on representative samples at different time points, which is essential for this study aims. Additionally, the availability of repeated information on fertility plans during the COVID pandemic on representative samples are quite rare, not only in Italy, highlighting a further unique contribution of our work. More specifically, we consider fertility intentions under two alternative formulations: 1) the classical short-term intention to conceive a child in the following 12 months, before and during the pandemic (i.e., in October 2016 and November 2020); 2) distinguishing between those who decided to confirm, postpone or abandon–at least temporarily– the pre-pandemic fertility plans in March and October 2020, among those retrospectively reporting to have a plan in January 2020 for conceiving a child in the same year.

The standard fertility intentions information allows to contrast them before and during the pandemic, and–potentially–with results from other surveys which use the same formulation. Thus, this step is introductive to the second part of our analysis, and it represents a possible bridge with the traditional way to analyse short-term fertility intentions in the demographic literature. The information about the decision to postpone or abandon the pre-pandemic fertility plan, instead, is quite innovative: while its non-standard formulation does not allow cross-studies comparisons, it offers the great advantage of possible insights on the mechanisms leading to fertility postponement, for which fertility behaviours would provide evidence only several years after the end of the pandemic. Therefore, the analyses conducted with the two formulations should be considered as complementary.

Another way in which our study is significantly contributing to the understanding of the possible impact of the COVID crisis on fertility plans is given by how we operationalized the economic uncertainty potentially arisen with the (threat of the) recession. In particular, by considering multiple indicators of the possible expected and experienced impact of the recession on the individual's financial and occupational uncertainty, we can explore complementary mechanisms through which the COVID pandemic can indirectly affect fertility intentions. Previous literature, in fact, highlights the special importance of subjective indicators of economic uncertainty (e.g., perceptions and expectations) when analysing its impact on demographic behaviours (see the impact of economic uncertainty on fertility intentions and behaviours section). Additionally, whether the fertility plans have been mostly affected by the experienced or expected uncertainty induced by the crisis might depend also by the phase of the COVID crisis. The availability of data collected at two time points in the 2020 allows exactly to distinguish between a very first impact of the crisis (in March, when the dread of a possible global recession and the shock of the first lockdown were the possible predominant drivers) to a later one (in October, when the recession signs were already evident, while the summer break had relieved the worries induced by the health emergency). Finally. gender differences are also explored: in contexts where traditional gender culture is still widespread, such as in Italy, economic recessions tend to negatively impact especially on men's side through an income effect ([27]: childbearing represents an additional economic risk, which should be avoided), while it is more common among women the rise of a substitution effect ([28]: in times of economic recession, investing in childbearing and care tasks can reduce the overall perceived uncertainty).

Working on the Italian case is also especially interesting, because of its peculiarity in the European context in terms of pandemic incidence and its consequent births decline, as well as for its pre-pandemic fertility levels and particularly unfavourable labour market characteristics. Italy, in fact, was the first Western country hit by the pandemic, with one of the highest relative number of cases and deaths due to the COVID-19 till now [29, 30]: because of the severe restrictive measures adopted to contain the spread of the virus, the economic shortcomings indirectly associated with the pandemic might be stronger than in other countries. Additionally, compared to other Western European countries, in Italy the economic and social consequences of the 2008 Great Recession have been stronger and more long-lasting, mainly hitting the young population. Women and young individuals have been the groups of workers mostly affected by this crisis in particular, and in general by the deregulation of the Italian labour market [31, 32]. In fact, in 2019 Italy showed one of the highest rates of youth unemployment (22.4% among young people aged 15–29 vs 11.4% in the EU27. Source: https://appsso.eurostat.ec.europa.eu/nui/show.do?dataset=yth_empl_100&lang = en) and the highest proportion of NEET–i.e., young people neither in employment nor in education—in Europe (23.8% among young people aged 15–34 vs 14.0% in the EU27. Source: https://appsso.eurostat.ec.europa.eu/nui/show.do?dataset=yth_empl_100&lang=en). Moreover, female labour force participation remains the lowest in Europe [33, 34], also because of a lack of accessible and affordable childcare services [35], which makes it difficult to reconciliate work and family obligations [36]. In this context, where occupational vulnerability is a widespread condition among young people, gender norms are not egalitarian and the welfare state is not supportive of families, fertility plans of individuals with poor economic prospects are particularly at risk [e.g., 37]. Italian studies have found that precarious job conditions are strongly related to fertility postponement [14, 38] and to reduced fertility intentions [39, 40]. Indeed, before the pandemic, Italy had the lowest fertility rate in the EU area (in 2019 was 1.27 in Italy vs 1.53 in the EU27. Source: http://appsso.eurostat.ec.europa.eu/nui/show.do?dataset=tps00199&lang=en). This also means that the concerns about the demographic future of one of most rapidly ageing

European country (the 2019 old-age dependency ratio–i.e., the number of persons aged 65 + per 100 persons aged 15–64 –was 35.8 in Italy vs 31.4 in the EU27. Source: https://ec.europa. eu/eurostat/databrowser/view/tps00198/default/table?lang=en) were already an issue before the COVID-19 pandemic [41].

As a consequence of the COVID-19 related recession, national statistics [42] report a drop in the percentage of employed individuals (-1.9 percentage points (pp) during the year 2020), largely due to fixed term contracts not being renewed. The freeze on layoffs, in fact, which was a strategy adopted by the Italian government for the entire 2020 at least for a number of sectors [43], has limited the impact of the recession on the (un)employment rates due to a possible increase of terminations of work contracts because of the crisis [44].

Also, the percentage of economically inactive people has increased, especially among women (+2 pp, versus +1 pp for men) and individuals aged 25–34 (+8.3 pp, against an average of 3.8 pp for the whole population). Some studies show that during the pandemic, Italians [45], and especially young Italians [46], expected a negative impact of the COVID crisis on their income, work and career. According to Luppi and Rosina [46], in October 2020, almost 45% of women and 35% of men expected future worse family income and work conditions because of the recession.

The experienced and the expected economic shocks derived by the COVID-19 related economic crisis may accelerate the Italian demographic recession because the negative consequences of the COVID-19 pandemic add to an already unfavorable context for fertility. Evidence from the late 2020 and the 2021 shows a strong decline in the Italian birth rate (especially from December 2020 to March 2021) [4], significantly associated with a pandemic effect [3]. Because fertility intentions are strong predictors of fertility behaviours [45–49], especially among younger individuals [50], our research sheds light on how the increased economic and occupational uncertainty due to the COVID crisis has compromised young Italians' fertility plans which are behind this birth rate decline.

## Economic recession and fertility intentions

With reference to past recessions [51, 52], it has been shown that childbearing has been more frequently postponed in those contexts where the crisis hit the most especially among childless young individuals [53–55]. Having children implies substantial financial efforts [56]; thus, job and income loss induces people to delay or withdraw their fertility plans [13, 57, 58]. Few studies have specifically focussed on the relationship between recessions and fertility intentions. Most of them focused on the effect of the Great Recession [16–18, 59], finding that perceived and experienced job and income worsening due to the crisis negatively influences fertility intentions. A short-term perspective on the consequence of a recession on fertility should primarily focus on intentions instead of behaviours, to better understand whether a short-term reduction in fertility is driven by a postponement or a possible withdrawal from the original plans.

## The impact of economic uncertainty on fertility intentions and behaviours

In periods of economic recessions, increased economic uncertainty has been shown to have detrimental consequences on fertility [60]. An enormous increase in economic uncertainty during the COVID-19 pandemic has already been widely documented [11, 61–64]. Economic uncertainty has been traditionally defined and measured through labour market indicators such as being unemployed or being employed with a temporary contract [12–14, 38, 39, 57, 65]. Usually, casual, project-based, and seasonal works imply a low labour market integration

and a wage penalty if compared with those employed with temporary or permanent contracts [66, 67]. Moreover, the former jobs tend to bend more physically and mentally tiring, with non-standard working hours, which makes work-family balance particularly difficult [68–70].

Evidence supports the existence of a negative effect of having a more precarious job on fertility intentions [15, 17, 71]. However, this strictly depends on contextual characteristics such as the structure of the labour market and the welfare system, which can act as institutional filters [72]. In a context of institutional and welfare state weakness, a condition of economic uncertainty is much likely to negatively impact on fertility decisions, as found by Novelli and colleagues [16] in Italy during the Great Recession.

An increasing number of studies on fertility intentions also considers the role of subjective indicators of economic uncertainty [12–14, 16, 73–75]. Economic and labour market deterioration is associated with an increase in perceived economic insecurity and occupational instability, and the individuals' perception and expectation may be of particularly relevance for fertility intentions on top of objective economic conditions [39, 40]. As highlighted in the literature (e.g., [18]), in fact, perceptions and expectations even more than the actual impact of the recession on individuals' current occupational and financial conditions can shape fertility plans. Thus, our study focuses also on the subjective dimensions of economic uncertainty, intended as individuals' expectations about their future income and occupational insecurity.

## The impact of the COVID-19 pandemic on births and fertility intentions

Studies on Italy and on other developed countries have shown mixed evidence of a possible fertility reduction (in intentions and behaviours) due to the pandemic [3, 4, 20–22, 46, 76–80], as in low and medium-income countries [81–83].

In Italy, data released by ISTAT in 2020 and 2021 [84] show that, compared to the same months of the previous year, in December 2020 the number of births declined by 10.3%; this is the first month in which the effects of the first epidemic wave are observable. A fluctuating trend in births, mostly declining, is observable also for the large part of the 2021. This trend has been anticipated by studies on fertility intentions conducted soon after the beginning of the health emergency in March 2020. In particular, Luppi and colleagues [79] show that at the beginning of the health emergency in March 2020, across European countries, a high proportion of young individuals of prime reproductive age (18–34 years old) were postponing or indefinitely suspending the original intention of having a child during the 2020. In particular, in Italy the percentage of those who abandoned (at least) temporarily their fertility plans for the 2020 was much higher than in France and Germany, and in general it was the highest in the group of countries included in the study. Another study conducted in the same period [80] estimated that 37% of the Italians aged 18–46 who were planning to have a child before the pandemic abandoned their plan, most of them because of worries about future financial difficulties brought by the recession. On the same line, Guetto and colleagues [76] found that pre-pandemic fertility intentions among Italians decrease with the increase of the expected length of the health emergency. All these studies have been conducted during the nationwide *lockdown* of March 2020.

Our study offers a twofold contribution to this preliminary evidence on the consequences of the pandemic on fertility. First, we examine the possible consequences of the economic uncertainty created by the pandemic on changes in fertility plans by studying the role of experienced and expected financial and occupational vulnerabilities due to the COVID-19 crisis in Italy. Second, our analyses are not limited to the very first period of the pandemic as most previous contributions, but we also exploit data collected in October and November 2020. Also,

we exploit pre-COVID data (2016) for the sake of comparing the pandemic context with a pre-pandemic period.

## Data and method

### Data

Data used for the analyses come from the Rapporto Giovani survey, carried out by the Osservatorio Giovani of Istituto Toniolo, in collaboration with IPSOS using CAWI (Computer Assisted Web Interviewing) administered questionnaires. The survey includes both regular and special cross-sectional waves: the regular module is repeated every year; the special modules are carried out as independent surveys on ad hoc topics. We exploit the COVID waves and a pre-COVID regular 2016 wave that is the most recent pre-pandemic wave including a fertility intention question that is comparable to the question included in the regular 2020 wave. In this way we exploit two different approaches to assess changes in fertility plans during the COVID-19 pandemic based on: 1) retrospective questions in the COVID waves; 2) repeated questions in October 2016 and November 2020 regular modules. Because retrospective questions can be biased, the 2016–2020 analysis can shed light–at least partially—on the interpretation of the results, eventually validating those from the COVID waves.

The two cross-sectional COVID waves have been administered on independent samples of individuals between March 27–31 and again in October 5–14, 2020. In all surveys, individuals are chosen with a quota sampling technique targeting the young population (18–34 years old individuals) ensuring the representativeness with respect to a significant set of key variables (gender, age, geographical origin, education, marital status, etc.) on which the quotas are defined (more information available at: https://www.rapportogiovani.it/osservatorio/). We select our working samples by excluding full-time students because we are not able to distinguish those who just started their academic studies from those who are about to exit the educational system. Additionally, we do not have information on whether they are actively looking for a job. This information would be relevant because students might give a very different meaning to the expectation about the impact of the current crisis on their future occupation and financial situation, depending on how close they are to entering the labour market. By excluding students, the sample sizes were 4573 (wave 2016), 4580 (wave 2020), 1491 (COVID-wave March 2020), and 1492 (COVID-wave October 2020). Then, the regular waves (2016 and 2020) have been pooled and only individuals with intentions to conceive within 3 or 2 years have been considered for the multivariate analyses (N = 3286). The COVID waves have also been pooled and only individuals who retrospectively declared to have planned to conceive a child in January 2020 have been selected (N = 758). More details on the samples' composition are available in the Results section. Because excluding full-time students we are considerably restricting our sample, as a robustness check, we re-run all the analyses by re-including full-time students (as a separated occupational category): results are always stable (results from the models with full-time students are now reported in the S1 Appendix, see S1_2-S1_6 Figs). Thus, excluding full-time students is not affecting our results: we can assume that, at least for the variables we are considering, the sample distributions are not significantly modified.

### Dependent variable based on the 2016 and 2020 regular waves

To compare the 2016 and 2020 fertility intentions we consider the short-term intentions collected through the question "Do you expect to conceive a child within the next 12 months?" with four possible answers: "Surely not", "Probably not", "Probably yes" and "Surely yes". The question has been asked only to individuals who answered that they intend to conceive a child in the following 3 years (in the 2016 survey) or 2 years (in the 2020 survey). This different time

frame might imply a possible overestimation of fertility intentions in 2016. However, these data provide a good measure on the short-term intention of conceiving a child. Although these questions are not strictly comparable because of the different time horizon, the sub-questions using a 1-year time frame are comparable and we focus on them.

### Dependent variable based on the COVID waves

In order to get information on respondents' fertility plans before the COVID-19 emergency, the following question has been asked:

a. "At the beginning of the year [2020], i.e., shortly before the coronavirus emergency broke out, were you planning to conceive/have a baby before the end of 2020?", using the possible alternatives [1a] "No", [2a] "I was considering it but without having planned it", [3a] "Yes, I had planned it". Only for the October wave a further possible answer has been added [4a] "Yes, I've both planned and realized it", for those who were able to conceive a child during 2020.

Respondents who declared to be planning to have a child during the 2020 (i.e., answers [2a] or [3a] in the March wave; answer [3a] in the October survey), have also been asked the following question:

b. "Did the coronavirus emergency interfere in any way with this plan?" with three possible answers: [1b] "No, the plan remains confirmed for 2020"; [2b] "The plan remains confirmed, but I had to postpone it"; [3b] "For now I have quit the plan".

Therefore, our second dependent variable is a three-level categorical variable corresponding to the following groups of individuals:

I. those who realized their plans [4a], those who were planning or were considering the idea to conceive/having a child during the 2020 and that confirmed their plan [1b] (i.e., *still-planners*);

II. those who were planning or were considering the idea to conceive/having a child during the 2020 and that postponed the plan [2b] (i.e., *postponers*);

III. those who were planning or were considering the idea to conceive/having a child during the 2020 and that have quit the plan [3b] (i.e., *abandoners*).

Notice that, obviously, in the October wave, those who reported to have already realized the original plan have no counterpart in the March wave and have been considered together with those who confirmed the original fertility plan. Also, the filter to the question on the confirmed and revised fertility plans changed between the two survey waves (see the two different filters above about the question [b]). In March, this question has been asked to both those who declared to have had strong [3a] and weaker pre-pandemic intentions [2a] to conceive a child during the next 12 months; in October, the question has been asked only to those with strong intentions [3a]. Because of this difference in the questionnaire design, we have been forced to take a decision about whether using all the respondents in March 2020 or considering only those with strong retrospective intentions, which is the sub-group fully comparable with the respondents of the October 2020 survey. In the following, we will focus on the "enlarged" sample that includes those with weaker intention in March 2020. However, we have considered a robustness check on the "restricted" sample that only includes those with stronger intentions both in March and October 2020 (see S1_2 Text and S1_7 Table in the S1 Appendix). Additionally, because there might be a selection among those who retrospectively declared to have fertility plans in January 2020, further analyses have been implemented by modelling the probability of being among those with no fertility plans in January 2020 (results are in the

S1 Appendix: see S1_1 Text and S1_6 Table). It might be the case, in fact, that those holding pre-COVID fertility plans (i.e., our analytical sample) represent a selected group characterized by more stable income and occupational conditions, while we expect economic uncertainty caused by the COVID crisis to be more relevant among those with no pre-pandemic fertility plans.

## Explanatory variables

The three main independent variables–which are indicators of the individual's occupational and financial uncertainty–are represented by: 1) occupational conditions; 2) the self-reported actual effect of the COVID-19 emergency on the respondent's financial situation; 3) the perceived expected effect on future personal income, and 4) occupation. While all the explanatory variables are available in the COVID waves, in the regular 2016 and 2020 survey only occupational condition is available.

The occupational condition of the respondents is operationalized in three categories, according to its vulnerability in the labour market, considering both the fact that they are working or not and the level of protection of their occupation and income:

I.  those who are not working and not studying (NEET): they represent the most vulnerable group in terms of personal income protection because they are out of the labour market. This category includes both individuals actively looking for a job, those who are available to work but they are not searching for a job (e.g., discouraged unemployed persons), and those who are not available to work at the moment of the interview (e.g., because of care responsibilities);

II. those who are working as self-employed (excluding professionals), precarious workers (such as those holding project-based contracts, causal workers, seasonal workers), and low-skilled autonomous workers: they are vulnerable both in terms of income and occupation protection. As already mentioned in the impact of economic uncertainty on fertility intentions and behaviours section, compared to employees with non-permanent contracts, they tend to have lower wages and consequently lower social security contributions. Also, they do not benefit from employment protection legislation [32];

III. those who are working as employees (with either permanent or non-permanent contracts), professionals and managers: they are those with the highest level of protection of both their income and occupation. Clustering together employees with permanent and non-permanent contracts is consistent with the evidence that the proportion of those still-planning, postponing or abandoning the fertility plan for the 2020 is similar between the two groups (i.e., tests of differences between proportions did not reject the null hypothesis of equality between the proportions of still-planning/postponing/abandoning individuals between the two groups of employees with and without a permanent contract). A further empirical confirmation of our clustering strategy is provided by performing the models as described in the "Method" section by including the occupation categories at the maximum level of disaggregation (for further details see S1_3 Text and S1_1 Fig in the S1 Appendix).

Regarding the self-reported effect of the crisis on the financial situation, the question has been posed as follows: "Compared to before the coronavirus emergency, your financial situation today: [1] is much worse; [2] is somewhat worse; [3] no change; [4] is somewhat better; [5] is much better". The variable has been dichotomized taking value 1 in the case the respondents report their financial situation as worsened and 0 otherwise.

As for the expected future effects, respondents answered to the question "Thinking about your future, do you think the current coronavirus emergency will have a positive or

negative impact on: your (personal) income / your occupation?" on a scale scoring from 1 (very negative) to 5 (very positive). The variable has been dichotomized, taking value 0 if the respondents expect a positive effect or null effect (values 3, 4 and 5), and value 1 in case the respondents expect negative returns in terms of income /occupation (values 1 and 2).

## Control variables

We control for the age of the respondent dividing the age range into 3 intervals (i.e., 18–24; 25–29; 30–34) to better catch possible life-cycle effects. We also control for marital status (marriage vs other), the presence of children in the household, and education (having or not a tertiary level degree). Unfortunately, there is no information on age and number of children. We also control for living in a region characterized by high level of diffusion of the COVID-19. In particular, our indicator is the cumulated number of confirmed cases of COVID-19 per 1000 inhabitants (Sources: https://github.com/open-covid-19/data#me tadata, https://www.data.gouv.fr/fr/reuses/carte-de-levolution-du-covid-en-france/, https:// coronavirus.data.gov.uk/#regions,http://www.salute.gov.it/portale/nuovocoronavirus/ dettaglioContenutiNuovoCoronavirus.jsp?area=nuovoCoronavirus&id=5351&lingua= italiano&menu=vuoto). By calculating the tertiles of the distribution, separately for the two survey waves, a control dummy variable for living in a region with high number of cumulated cases of COVID-19 has been added (regions above the second tertile). Both in March and in October, these regions were: Valle d'Aosta, Emilia-Romagna, Liguria, Lombardia, Piemonte, Trentino Alto-Adige and Veneto. As previously mentioned, the COVID crisis has not just economic side-effects: the seriousness of the health emergency and the subsequent social adjustment can have played a role in shaping fertility plans too.

## Method

We implement two distinct modelling strategies to contrast fertility intentions before and after the pandemic by exploiting data in the regular waves (2016–2020) and in the COVID waves (March and October 2020). Given the ordered nature of the dependent variable measuring fertility intentions available in the regular waves, we use generalized ordered logit models by pooling the two samples and including the occupational status–interacted with the survey year–as the main explanatory variable.

Instead, the categories of the second dependent variable on whether pre-pandemic fertility plans have been confirmed, postponed, or abandoned do not follow a natural order. Therefore, data from the COVID waves, have been analysed using multinomial models to assess the separate effects of explanatory variables on each category of the outcome. The analyses are conducted by pooling the March and October waves and controlling for the timing of the survey (reference: March 2020). We first run a model that includes all control variables and the occupational conditions. Then, additional models add in turn one of the subjective indicators of economic uncertainty. We also tested interactions between the main covariates and the wave indicator but did not find statistically significant results except for the occupational conditions. Therefore, while for the subjective indicators of economic uncertainty results are presented without including the interaction with the wave, for the occupational conditions the results for the interaction terms are shown. To ease interpretation of findings, all results are presented graphically by showing predicted probabilities. Weights have been always applied in both descriptive statistics and models. Full tables of estimated coefficients are available in the S1 Appendix (S1_1-S1_5 Tables).

**Table 1. Samples' size and distribution of the dependent variables, main covariates and control variables for the regular waves (2016 and 2020); separated and pooled samples.**

|  | Regular waves | | |
|---|---|---|---|
|  | **2016** | **2020** | **Pooled** |
| **Original sample size** | 6172 | 7012 | 13184 |
| **Sample size without students** | 4573 | 4580 | 9153 |
| **Intention to conceive a child within 2/3 years** |  |  |  |
| Yes (%) | 39.78 | 32.03 | 35.90 |
|  | 1819 | 1467 | 3286 |
| **Intention to conceive a child within 12 months[a] (%)** |  |  |  |
| Surely not | 21.94 | 17.66 | 20.02 |
| Probably not | 41.62 | 39.20 | 40.54 |
| Probably yes | 26.61 | 33.61 | 29.73 |
| Surely yes | 9.84 | 9.54 | 9.71 |

Notes:

[a] only for those with 2/3 years fertility intentions

# Results

## Descriptive statistics

In Tables 1 and 2 we report the samples sizes and the distribution of the dependent variables in the regular waves (2016 and 2020) and in the COVID waves (March and October 2020), respectively.

**Table 2. Samples' size and distribution of the dependent variables, main covariates and control variables for the COVID waves (March and October 2020); separated and pooled samples.**

|  | COVID waves | | |
|---|---|---|---|
|  | **March** | **October** | **Pooled** |
| **Original sample size** | 2000 | 2000 | 4000 |
| **Sample size without students** | 1491 | 1492 | 2983 |
| **Retrospective fertility plans in January 2020 (%)** |  |  |  |
| With no fertility plans (*no-plans*) | 70.10 | 65.20 | 67.68 |
| With some possible plans | 18.90 | 11.20 | 12.54 |
| With a precise plan | 11.00 | 16.70 | 13.91 |
| With a plan that has been realized | - | 6.90 | 3.45 |
| **Sample size without no-plans[a]** |  |  |  |
| enlarged sample[b] | 445 | 313 | 758 |
| restricted sample[c] | 165 | 313 | 478 |
| **Revised/confirmed fertility plans in the enlarged sample (%)** |  |  |  |
| Still-planners (considering also the realized plans in October) | 28.54 | 45.69 | 35.62 |
| Postponers | 38.00 | 37.38 | 37.99 |
| Abandoners | 33.03 | 16.93 | 26.39 |
| **Revised/confirmed fertility plans in the restricted sample (%)** |  |  |  |
| Still-planners (considering also the realized plans in October) | 48.49 | 45.69 | 47.13 |
| Postponers | 27.27 | 37.38 | 32.83 |
| Abandoners | 24.24 | 16.93 | 20.04 |

Notes:

[a] considering only those answering the question regarding the revised/confirmed fertility plans

[b] considering also those "with some possible plans" in March 2020

[c] excluding those "with some possible plans" in March 2020

As for the regular waves, Table 1 shows that in 2016, almost 40% of the respondents does intend to have a child in the following 3 years; in 2020 the percentage of individuals who intend to have a child in 2 years is about 32%. Among those with a positive 3 or 2 years fertility intention, in 2016, about 36% intends to have a child within in short-term (i.e., 12 months; those answering probably or surely yes), while in 2020 this percentage is about 43%. If we calculate these percentages on the total samples (excluding students) we get very similar values: 14.5% and 13.8%, respectively. Thus, the positive fertility intentions within 1-year are very similar in 2016 and 2020.

Data from the pooled COVID waves (Table 2) show that almost 68% of the respondents declare that they did not intend to conceive a child in the next 12 months in January 2020. Among the remaining 32%, the percentage of those confirming, postponing, or abandoning the fertility plans because of the pandemic occurrence has been calculated both in the restricted and the enlarged samples. To check the comparability of the two samples, and only for this descriptive aim, we compare the restricted samples of March and October contrasting groups of respondents selected with the same criteria (i.e., including only those retrospectively declaring to have a clear fertility plan in January 2020). In the restricted samples we find that the distribution of changes in fertility plans is similar in the two periods: in October, about 45.7% of the respondents (48.5% in March) confirms the original plan of conceiving a child during the 2020. The proportions of postponers and abandoners slightly vary between the two surveys, with a little increase in the percentage of postponers (from about 27.3% to 37.4%) and a correspondent decline for the abandoners (from about 24.2% to 16.9%).

Tables 3 and 4 report the distribution of the main covariates and control variables, respectively in the samples of the regular surveys and in the samples of the COVID surveys.

## Comparing the 2016 and 2020 fertility intentions in the regular waves

Although we do not observe substantial differences in fertility intentions between the 2016 and 2020 samples as it might be expected because of the 2020 COVID-19 emergency, a variation can be found when looking at the occupational condition of the respondents. In particular, by running a generalized ordered logit model for the intention to conceive a child in the following 12 months by gender, we contrast the predicted probabilities of fertility intentions by occupation in 2016 and 2020 (complete estimates are in S1 Appendix, S1_1 Table). Fig 1 reports

**Table 3. Distribution of the main covariates and control variables for the regular waves (2016 and 2020); separated and pooled samples.**

| | Regular waves | | |
|---|---|---|---|
| | **2016** | **2020** | **Pooled** |
| Occupational status | | | |
| *NEET* | 24.47% | 21.79% | 23.13% |
| *Professionals/employee* | 65.58% | 66.76% | 66.17% |
| *Self employed/temporary workers* | 9.95% | 11.45% | 10.70% |
| Women | 55.17% | 57.13% | 56.15% |
| Age class | | | |
| *18–24* | 8.44% | 9.10% | 8.77% |
| *25–29* | 33.21% | 32.45% | 32.83% |
| *30–34* | 58.35% | 58.44% | 58.40% |
| Tertiary education | 22.57% | 24.45% | 23.50% |
| Cohabiting/married | 35.85% | 45.62% | 40.72% |
| With children | 25.74% | 27.24% | 26.49% |
| ***Sample size*** | *1819* | *1467* | *3286* |

**Table 4. Distribution of the main covariates and control variables for the COVID waves (March and October 2020); separated and pooled samples.**

| | COVID waves | | |
|---|---|---|---|
| | **March** | **October** | **Pooled** |
| Occupational status | | | |
| *NEET* | 23.60% | 12.80% | 18.80% |
| *Professionals/employee* | 62.00% | 71.80% | 66.41% |
| *Self employed/temporary workers* | 14.40% | 15.40% | 14.79% |
| Financial situation has worsened (a) | 50.84% | 29.33% | 41.48% |
| Perceived income at risk (b) | 51.32% | 24.17% | 39.51% |
| Perceived occupation at risk (b) | 54.79% | 73.16% | 62.78% |
| Women | 49.10% | 53.30% | 51.00% |
| Age class | | | |
| *18–24* | 21.28% | 18.56% | 20.09% |
| *25–29* | 27.10% | 27.43% | 27.24% |
| *30–34* | 51.60% | 54.01% | 52.66% |
| Tertiary education | 23.98% | 18.83% | 21.74% |
| Cohabiting/married | 48.01% | 53.12% | 50.23% |
| With children | 27.38% | 40.62% | 33.14% |
| *Sample size* | *445* | *313* | *758* |

(a) "Compared to before the coronavirus emergency, your financial situation today: [1] has worsened a lot; [2] has slightly worsened; [3] nothing has changed; [4] has slightly improved; [5] has improved a lot". The variable has been dichotomized taking value 1 in the case the respondent reports their financial situation as worsened and 0 otherwise.
(b) "Thinking about your future, do you think the current coronavirus emergency will have a positive or negative impact on: your (personal) income / your occupation?" on a scale scoring from 1 (much negative) to 5 (much positive). The variable has been dichotomized, taking value 0 if the respondent expects a positive effect or null effect (values 3, 4 and 5), and value 1 in case the respondent is expecting negative returns in terms of income /occupation (values 1 and 2).

results from generalised ordered logistic regression models. Instead of representing coefficients (on the log-odds scale) which compare one category with the previous, we estimated predicted probabilities of each outcome's category by the values of the explanatory variables. However, instead of reporting all predicted probabilities, we only show those for which a statistically significant difference across the groups defined by the explanatory variables is found. More specifically, because we do not find significant variations for the other categories of the outcome variable, Fig 1 shows the predicted probabilities corresponding to the extreme category "surely yes". The figure shows that in 2020 most of the vulnerable workers (i.e., self-employed and temporary workers) reports a lower probability of definitively intending to have a child in the short-term as compared to their counterparts in 2016. This pattern is evident for both genders, and it is statistically significant ($p<0.05$) for women (with a predicted probability of 16% in 2016 vs 3% in 2020) and for men ($p<0.10$ with a predicted probability of 17% in 2016 vs 5% in 2020). On the contrary, for those with a safer employment situation the probability of intending to have a child "for sure" within the 12 months has not significantly (neither statistically not substantially) changed for men, and for women it is even slightly higher in 2020 than in 2016 (respectively 16% vs 11%, $p<0.05$).

## Revised fertility plans in the COVID waves

We now turn to data from the COVID surveys that allow exploring self-reported changes in pre-COVID fertility plans. We first examine whether the occupational conditions are

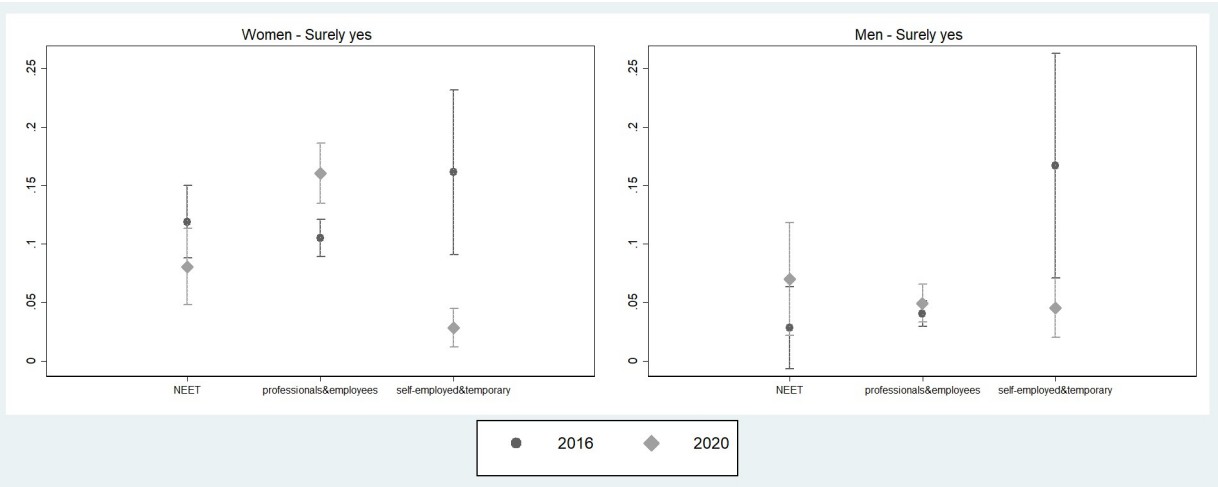

**Fig 1. Predicted probabilities of intending to have a(nother) child (answer: "Surely yes") in the next 12 months by occupational condition in 2016 and 2020 and by gender (confidence intervals for pair-wise comparisons at the 10% significance level).** Note: Predicted probabilities bases on a generalized ordinal logit model, controlling for age class, education, marital status, presence of children, among those intending to have a child in the next 2 years (2020) or 3 years (2016). Full results are available in S1_1 Table in S1 Appendix. Confidence intervals are calculated to allow for multiple comparisons of predicted probabilities, i.e., to allow testing their statistical equality at an approximate 5% significance level. Overlap between a pair of intervals indicate that the corresponding predicted probabilities are not statistically different at the 10% level, while non-overlap indicate a significant difference. When differences of interest are significant at the 5% level, this is noticed in the text.

associated with the probability to confirm, postpone, or abandon the reported pre-COVID fertility, by gender (Fig 2; Model 1 in S1_4 and S1_5 Tables in the S1 Appendix). Fig 2 shows that in March a more precarious occupational condition is associated (p<0.05) with a higher chance of abandoning the original plan compared to October. For women, in March the probability of abandoning the fertility plans is about 46% for self-employed and temporary workers and 43% for NEET (vs 29% for employees). Correspondently, a lower probability to confirm the pre-COVID plans arises in March (19% for self-employed and temporary workers, 26% for NEET vs 36% for employees). In October the probability for women to abandon the fertility plan is 16% for both the vulnerable occupation categories, and it is no longer significantly different from the same probability for the employees. For men, the chances of abandoning the plan for self-employed and temporary workers are significantly higher in March (54%) than in October (16%) and compared to the other two occupational categories in March (NEET 28% and employees 23%), while the difference is no longer significant in October. Finally, the probability to confirm the fertility plan is lower for self-employed and temporary workers in March (9%) compared to October (38%) and to NEET (26%) and employees (33%) in March.

Figs 3–5 show the results from models where the subjective indicators of economic uncertainty have been added one by one; as explained in the "Method" section, the two COVID waves have been pooled in this case (complete estimates are in S1_2 and S1_3 Tables in the S1 Appendix). For both genders, but especially for women, Fig 3 shows an association between having experienced a drop in the income and the changes in fertility plans (p<0.05). In particular, individuals who report a worsened financial situation display a higher probability to abandon their fertility plans for the 2020 as compared to individuals who did not experience such an income shock (respectively 36% vs 21% among women and 30% vs 20% among men). Among women, a worsened financial situation is also negatively associated with the probability of confirming the fertility plan (32% for those who did experience worsened financial situation vs 43% for those who did not).

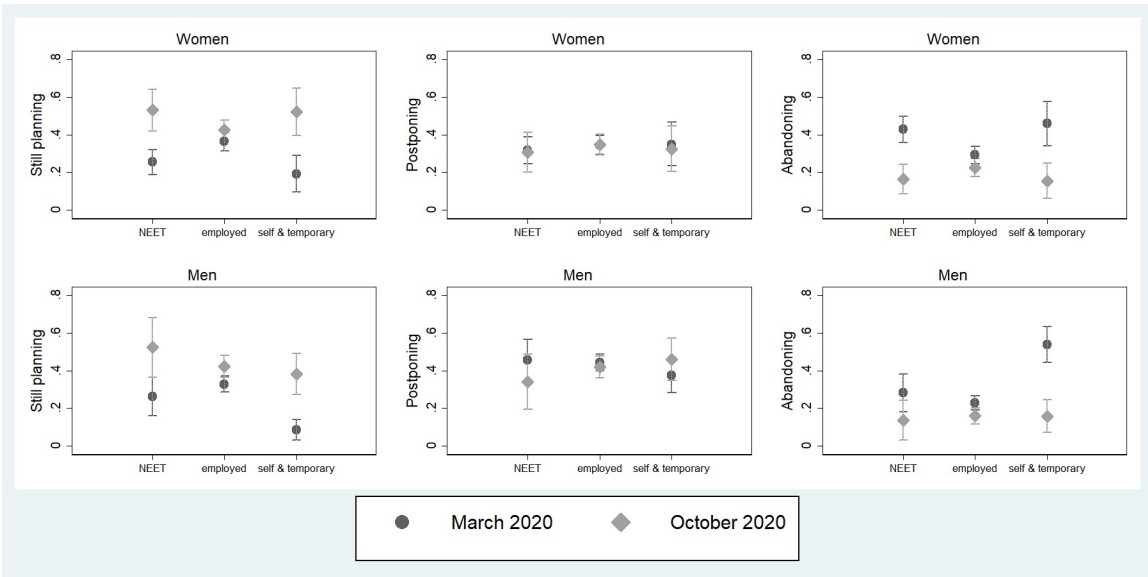

**Fig 2. Predicted probabilities of changes or confirmations of pre-COVID fertility plans in March and October 2020 across occupational conditions (confidence intervals for pair-wise comparisons at the 10% significance level).** Note: Predicted probabilities bases on a multinomial logit model, controlling for age class, education, marital status, presence of children. Full results are available in S1_4 and S1_5 Tables in S1 Appendix. Confidence intervals are calculated to allow for multiple comparisons of predicted probabilities, i.e., to allow testing their statistical equality at an approximate 5% significance level. Overlap between a pair of intervals indicate that the corresponding predicted probabilities are not statistically different at the 10% level, while non-overlap indicate a significant difference.

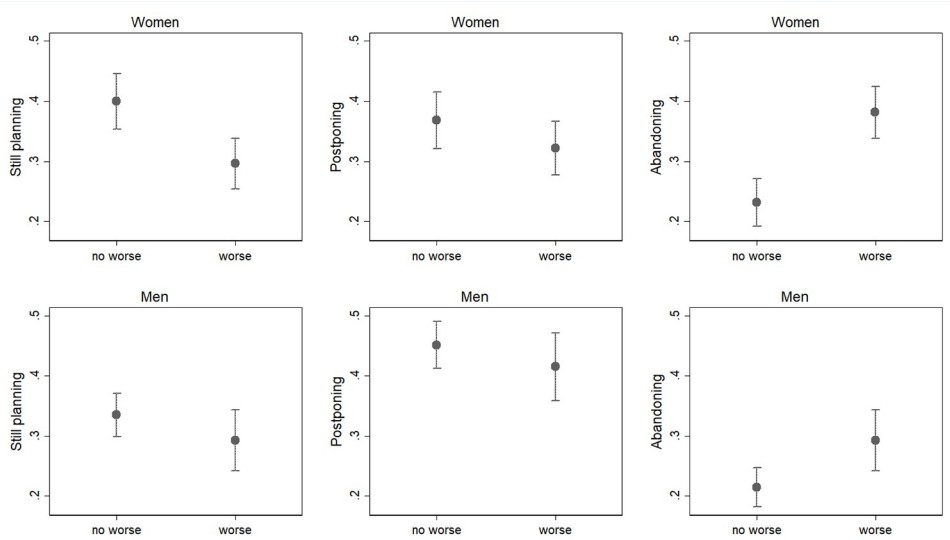

**Fig 3. Predicted probabilities of changes or confirmations of pre-COVID fertility plans in March and October 2020 for those who experienced and those who did not experience a worsening in their financial situation (confidence intervals for pair-wise comparisons at the 5% significance level; March and October 2020 waves pooled).** Note: Predicted probabilities bases on a multinomial logit model, controlling for age class, education, marital status, presence of children. Full results are available in S1_2 and S1_3 Tables in S1 Appendix. Confidence intervals are calculated to allow for multiple comparisons of predicted probabilities, i.e. to allow testing their statistical equality at an approximate 5% significance level. Overlap between a pair of intervals indicate that the corresponding predicted probabilities are not statistically different at the 10% level, while non-overlap indicate a significant difference. No differences in terms of significancy has been found with confidence intervals at 10%.

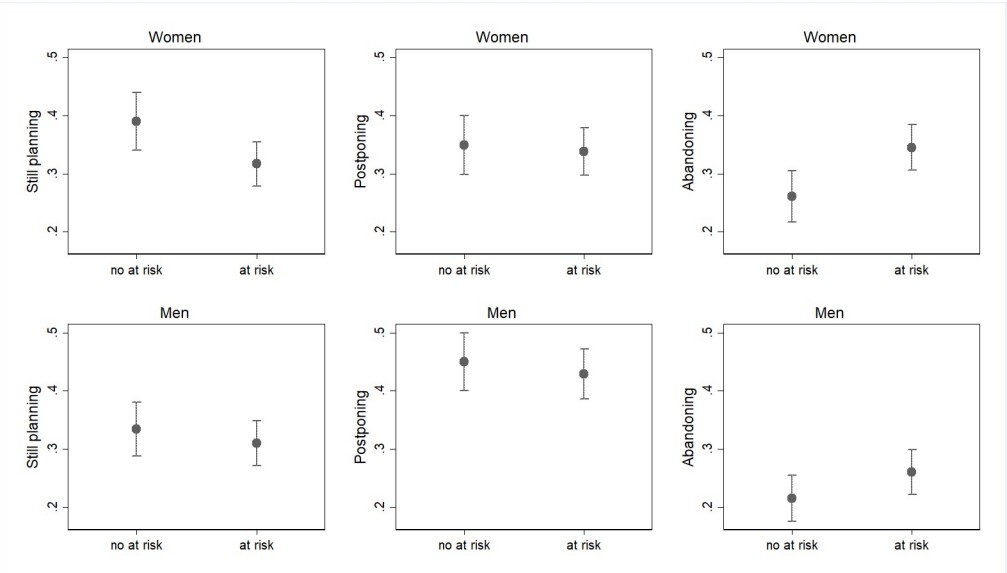

**Fig 4. Predicted probabilities of changes or confirmations of pre-COVID fertility plans in March and October 2020 for those who perceived and those who do not perceived their future occupation as at risk (confidence intervals for pair-wise comparisons at the 5% significance level; March and October 2020 waves pooled).** Note: Predicted probabilities bases on a multinomial logit model, controlling for age class, education, marital status, presence of children. Full results are available in S1_2 and S1_3 Tables in S1 Appendix. Confidence intervals are calculated to allow for multiple comparisons of predicted probabilities, i.e. to allow testing their statistical equality at an approximate 5% significance level. Overlap between a pair of intervals indicate that the corresponding predicted probabilities are not statistically different at the 10% level, while non-overlap indicate a significant difference. No differences in terms of significance has been found with confidence intervals at 10%.

Postponers do not show any significant pattern, with a slightly–even though not signifi-cant–similar behaviour to that of the still-planners.

Finally, Figs 4 and 5 present results regarding the expectations about the future effect of the recession on the individual's occupation and income, respectively (p<0.05). Women who per-ceive their occupation as at risk because of the COVID-19 crisis are less likely to confirm their pre-crisis fertility plans as compared to their counterparts who do not hold such a negative expectation (respectively 34% vs 42%, Fig 4). Correspondingly, women who expect their occu-pation to be at risk show a considerably higher probability of declaring to abandon their origi-nal fertility plans for the 2020 (respectively 32% vs 24%, Fig 4). Similarly, both women and men who perceive their future income to be at risk due to the COVID-19 crisis show a lower probability of confirming and a higher probability of abandoning the pre-pandemic fertility plan (Fig 5). More specifically, the probability of abandoning the pre-COVID fertility plan for women who expect the pandemic to produce a negative shock on their income in the future is 36% against 23% for those who do not hold such a negative expectation. For men, we observe a slightly lower but still sizeable gap (29% vs 20%).

As for the previous case, postponers do not show significant results, but again they tend to be similar to the still-planners.

## Discussion

A wide interest on the consequences of the COVID-19 emergency on fertility has arisen world-wide in the scientific community since the beginning of the pandemic. Prediction about drops in births have been largely shared among demographers (e.g., [78–80, 85–87]) and preliminary

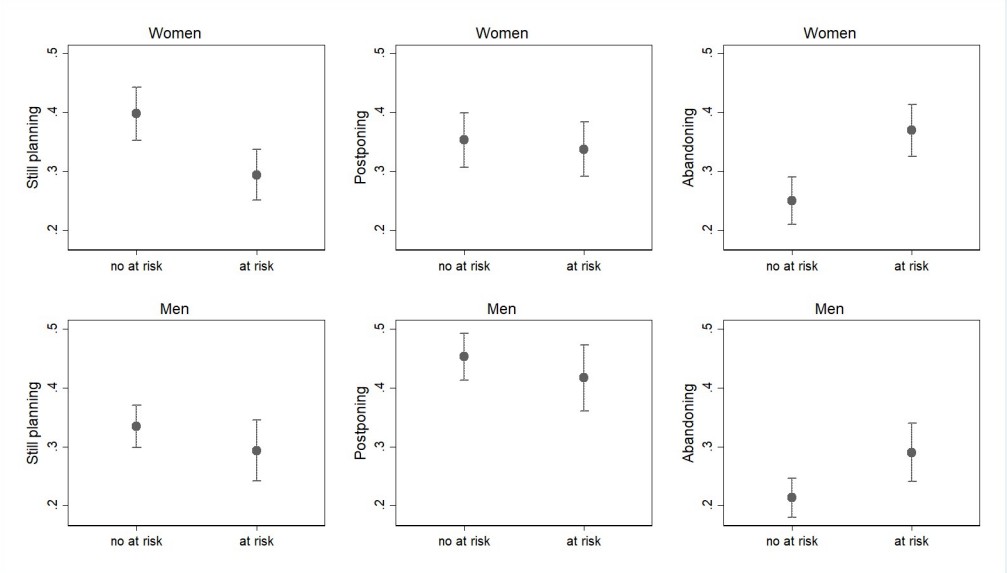

**Fig 5. Predicted probabilities of changes or confirmations of pre-COVID fertility plans in March and October 2020 for those who perceived and those who do not perceived their future income as at risk (confidence intervals for pair-wise comparisons at the 5% significance level; March and October 2020 wave pooled).** Note: Predicted probabilities bases on a multinomial logit model, controlling for age class, education, marital status, presence of children. Full results are available in S1_2 and S1_3 Tables in S1 Appendix. Confidence intervals are calculated to allow for multiple comparisons of predicted probabilities, i.e., to allow testing their statistical equality at an approximate 5% significance level. Overlap between a pair of intervals indicate that the corresponding predicted probabilities are not statistically different at the 10% level, while non-overlap indicate a significant difference. No differences in terms of significance has been found with confidence intervals at 10%.

data from birth statistics confirm a drop in birth rates 8–9 months after the beginning of the health emergency at least in Italy [3, 4, 20], as well as in the US and other European and Asian countries [21, 22]. In Italy, the very low fertility level was already a concern for demographers and policy makers before the pandemic. Hence, a possible further decline in fertility due to the COVID-19 health and economic crisis would posit new challenges to the holding of both the economic development and the welfare state in the medium term.

Despite a possibly greater interest on fertility behaviours when examining the demographic consequence of economic and health crises, fertility intentions are highly informative in the short-term, in part because changes in fertility behaviours need longer time to be detected and studies on micro-level data on births during the pandemic are still rare, but also because by investigating the intentions it is possible to consider in which way people are revising their fertility plans [18]. This represents one of the strengths of our study. In fact, we examine fertility intentions in two alternative ways: 1) using the classical intention to conceive a child in the following 12 months, before and during the pandemic; 2) distinguishing among confirmed, postponed or abandoned (i.e., indefinitely suspended) fertility plans for those who, before the pandemic, were planning to conceive a child in 2020. In particular, we were especially able to distinguish those who are postponing their pre-COVID plan from those who abandoned it because of the pandemic. This is important because we may expect that postponers will more likely delay the childbirth to a next future, while for the abandoners both the scenarios of a long-term delay and of a withdrawal are plausible. This last group of individuals can be considered particularly vulnerable, as their fertility plans seem to be more easily compromised by the crisis. This interpretation is supported by our results, which show a clear association between

higher levels of economic uncertainty and higher chances to abandon the fertility plan, while no clear results appear for postponers, which in general seem to behave closer to still-planners.

Additionally, it is worth to be noticed that the interest in studying the association between economic uncertainty and fertility intentions is confirmed by a number of recent studies that still analyse this relationship in the context of the Great Recession (e.g., [16, 88]. Even though data on fertility behaviours are largely available for those years, fertility intentions are more informative about the mechanisms behind the decision to conceive, as they represent the very first reaction to–for example as in this case–an economic shock.

From a methodological point of view, using two alternative approaches to examine the possible effect of COVID-19 on fertility intentions has strengthened the validity of our findings. In fact, we exploited both independent samples drawn before (2016) and during the pandemic (2020; "regular waves") and retrospective information collected during the pandemic on pre-COVID fertility plans ("COVID waves" collected at the beginning of the health emergency in Italy—March 2020—and at the beginning of the second wave of the epidemic—October 2020).

In Italy, occupational categories have been hit differently by the economic crisis, with self-employed and temporary workers much more affected than employees, at least in the 2020 [45]. Consistently with this, by comparing the short-term fertility intentions (within 12 months) among similar samples of young individuals in 2016 and 2020, we find that individuals with more vulnerable occupations show a lower probability of definitely intending to have a(nother) child soon. For self-employed and temporary workers, changes in fertility intentions between the two considered time points are considerable from a substantive point of view: from 16% to 3% for women, and from 17% to 5% for men. No significant differences appear among NEETs. This result contrasts the often reported finding that women invest more in childbearing during periods of economic uncertainty to substitute the loss of employment opportunities [89–92]. This might suggest an increasing similarity between women and men behaviours among the young generations, or at least in this specific age group. During the COVID-19 pandemic, in fact, among women the income effect (i.e., economic uncertainty inhibiting the demand for children) exceeds the substitution effect (i.e., economic uncertainty facilitating childbearing as unemployment increases the time for childcare). However, previous studies also show that the substitution effect prevails when the perceived employment and income fragility is perceived as a temporary condition; on the contrary, the income effect might be stronger when the perceived and experienced "unemployment trap" lasts longer [89, 93, 94]. Therefore, this might suggest a "scare-effect" [66] and possible long-term negative consequences of the COVID-crisis on the youth labour market in Italy.

To better understand the role of economic uncertainty brought by the COVID-19 pandemic and given that the COVID-19 pandemic is not egalitarian in its effects, we compared those who decided to indefinitely suspend their fertility plans because of the pandemic to those who postponed or confirmed the original fertility plan, by looking at both objective and subjective indicators of economic uncertainty. Consistently with results from other studies [14], women more than men tend to abandon the fertility plans for the 2020 when they experience or perceive an occupational and income uncertainty. In particular, we found that individuals who are more vulnerable in terms of occupational and financial conditions are more prone to indefinitely suspend the plan to conceive a child in the short term. However, differences across occupational categories appear only for data collected at the beginning of the health emergency–in March 2020 –while in October of the same year such differences are no longer present. This suggests that, at least before the second pandemic wave, the impact of the crisis on fertility plans tends to become more homogeneous across occupational groups, while the specific individual economic conditions (having or not being economically affected by the crisis) and expectations are equally relevant in both periods as our findings show.

One of the advantage of having different objective and subjective indicators of economic uncertainty is that this allows to overcome some of the single-indicator limits. For example, our data does not allow to consider variation in the occupational condition during the recession period, which can complicate the interpretation of our results at least for data collected in October 2020 (where differences across occupational conditions actually disappear). However, by exploiting the information about the loss of income during the pandemic period as an alternative indicator of economic uncertainty we partially overcome this limit. Actually, we found that the loss of income is associated with higher chances to abandon the pre-COVID plan.

The (qualitatively) different results we find for occupational conditions using the regular waves and the COVID-waves might be also partially related with the specific timing of the waves. The first COVID-wave was carried out in March 2020 during a strict lockdown where most shops and many economic activities were closed. Data for the second COVID-wave, instead, were collected in October after the relatively quiet summer, for what the pandemic dynamic is concerned, and in a period where more economic activities were operating compared to March. The regular 2020 wave was implemented in November, which can be considered both in terms of pandemic and economic conditions as in-between the March and October 2020 periods. Future studies, with longer periods of observation, might focus on the fertility intentions trends following the development of the health crisis, for example considering a possible impact of the vaccination campaign.

An additional special remark should be made about the results for the postponers category. The fact that they hardly show significant patterns might be related to the fact that this group could be quite mixed, with someone postponing for few months and then realizing the plan, while others intending to postpone longer. At the same time, the fact that postponers seem to be more similar to still-planners than to the abandoners might be because those who abandoned the plans adopted a very extreme position, also compared to the long-term postponers.

Because data are cross-sectional, we are not able to avoid possible selection (i.e., into a specific occupational category during the crisis) and endogeneity effects: for this reason, the aim of this study is merely descriptive and explorative. Longitudinal data would allow in the future to shed better light on possible mechanisms behind our results. This will also allow to overcome issues related with the non-homogeneous way in which some of our dependent and independent variables have been operationalized. A further limitation of our data is that they are restricted to young individuals aged 18–34. Although this is one of the groups that most likely experienced the negative economic consequences of the pandemic in Italy [45, 95], in case older individuals' objective or subjective economic circumstances have been affected the consequences on their fertility plans might be even stronger [96]. Additionally, because of the data constrain, we only focused on the negative impact of the pandemic on fertility plans; in other words, we did not consider those who did not have a plan to conceive a child in the pre-pandemic period but changed their mind during the health emergency. Micelli and colleagues [80] in fact, report a small but significant proportion of individuals who revealed a new desire for parenthood during the first lockdown (11.5%), and the 4.3% of them also tried to conceive a child during that period because, for example, they were valuing more their family investment and the time spent with the partner/children. There might be some cases in which the teleworking opportunities, spread by the epidemic, had favoured the work-family reconciliation, even though the average net effect of the health emergency was detrimental on the work-family balance, especially on the women's side [97]. Unfortunately, we do not have additional information regarding the job characteristics (e.g., public-private sector, use of teleworking, etc.) or the income level, which might help in better disentangling the association between fertility plans and the occupational conditions.

Despite the limitations, our findings have important implications. Fertility realizations to be observed need longer time span, especially if we aim to find the characteristics associated with higher chances of postponement and withdrawal from the original fertility plans. For this reason, because fertility intentions in our study have been operationalized exactly as the intention to confirm, postpone, or abandon the original fertility plan, we have a more precise idea on the level of vulnerability associated with fertility intentions during the early phases of the crisis. Our findings clearly point to the fact that the unequal economic (experienced or expected) consequences of the pandemic also produced—and will produce—heterogeneous effects on fertility. Despite the difficulties in limiting the negative effects of the economic recession on fertility, evidence of a decline in birth rates are especially present in countries where fertility where already below the European average. This means that, especially in these contexts, policy makers who aim at contrasting the possibly persistent COVID shocks on fertility should implement labor market and family policies that allow the individuals to plan irreversible important childbearing choices with a less uncertain horizon. In this respect, the 2021 Italian universal child allowance (AUUF–*"Assegno Unico e Universale per i Figli"*) has been adopted with the aim of creating more favourable conditions for childbearing; the allowance supports the financial investment for rearing children from the 7th month of pregnancy to 21 years, it is only partially income-based (i.e., there is a minimum which is universally provided) and it increases with the second child. This income-support policy should represent the starting point of the implementation of the so-called "Family Act", which aims at improving policies and tools for work-family reconciliation and childcare services. This policy intervention follows a similar set of measures that have been adopted in Germany starting from 2007; despite the country economy and labour market were highly performant, the German fertility rate was among the lowest in Europe. The adoption of a universal allowance for children, together with the empowerment of the parental leave system and the childcare services, has strongly pushed fertility rate upwards [98]. Because of the expected–and partially already experienced–negative effect of the COVID-19 recession on births in some developed countries, those countries that were already experiencing pre-pandemic downward fertility trends are now in the urgent need to create more favourable conditions for childbearing, avoiding that the postponement due to the recession would end into a further fertility decline.

## Supporting information

**S1 Appendix.** S1_1 Table. Ordered logit model for the fertility intentions at 12 months, by gender (Regular waves, 2016 and 2020). S1_2 Table. Multinomial models for the intention of still-planning, postponing or abandoning the pre-pandemic fertility plan, for women (COVID waves, March and October 2020). S1_3 Table. Multinomial models for the intention of still-planning, postponing or abandoning the pre-pandemic fertility plan, for men (COVID waves, March and October 2020). S1_4 Table. Multinomial models for the intention of still-planning, postponing or abandoning the pre-pandemic fertility plan with survey wave as mediator, for women (COVID waves, March and October 2020). S1_5 Table. Multinomial models for the intention of still-planning, postponing or abandoning the pre-pandemic fertility plan with survey wave as mediator, for men (COVID waves, March and October 2020). S1_1 Text. The selection into the "no plans" group. S1_6 Table. Logit models for the probability of not planning to have a child in the 2020, by including variables on income and occupation vulnerability due to COVID-19, in 2020 (Data source: Rapporto Giovani. March–October 2020). Robustness checks: S1_2 Text. Stronger and weaker retrospective fertility intentions. S1_7 Table. Multinomial models for the intention of still-planning, postponing or abandoning the pre-pandemic fertility plan, in the restricted and the enlarged samples (Data source: Rapporto

Giovani COVID survey, March 2020 and October 2020). S1_3 Text. Clustering occupational categories. S1_1 Fig. Predicted probabilities of changes or confirmations of pre-COVID fertility plans in March and October 2020 across occupational conditions (confidence intervals for pair-wise comparisons at the 10% significance level). Including full-time students in the analysed samples. S1_2 Fig. Predicted probabilities of intending to have a(nother) child (answer: "surely yes") in the next 12 months by occupational condition in 2016 and 2020 and by gender, including full-time students (confidence intervals for pair-wise comparisons at the 10% significance level). S1_3 Fig. Predicted probabilities of changes or confirmations of pre-COVID fertility plans in March and October 2020 across occupational conditions, including full time students (confidence intervals for pair-wise comparisons at the 10% significance level). S1_4 Fig. Predicted probabilities of changes or confirmations of pre-COVID fertility plans in March and October 2020 for those who experienced and those who did not experience a worsening in their financial situation, including full-time students (confidence intervals for pair-wise comparisons at the 5% significance level; March and October 2020 waves pooled). S1_5 Fig. Predicted probabilities of changes or confirmations of pre-COVID fertility plans in March and October 2020 for those who perceived and those who do not perceived their future occupation as at risk, including full-time students (confidence intervals for pair-wise comparisons at the 5% significance level; March and October 2020 waves pooled). S1_6 Fig. Predicted probabilities of changes or confirmations of pre-COVID fertility plans in March and October 2020 for those who perceived and those who do not perceived their future income as at risk, including full-time students (confidence intervals for pair-wise comparisons at the 5% significance level; March and October 2020 wave pooled).
(DOCX)

**S1 Dataset. Dataset covid survey 2020.** It is the minimal dataset (.dta format) including the variables used for the analyses on the COVID 2020 surveys.
(DTA)

**S2 Dataset. Dataset regular survey 2016–20.** It is the minimal dataset (.dta format) including the variables used for the analyses on the regular 2016 and 2020 surveys.
(DTA)

## Author Contributions

**Conceptualization:** Francesca Luppi, Bruno Arpino, Alessandro Rosina.

**Formal analysis:** Francesca Luppi.

**Methodology:** Bruno Arpino.

**Project administration:** Alessandro Rosina.

**Writing – original draft:** Francesca Luppi, Bruno Arpino, Alessandro Rosina.

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
