## [Decision Letter · Decision Letter 0]

21 Apr 2022

PONE-D-22-07293Fertility plans during the COVID-19 pandemic in Italy: the role of occupation and income vulnerabilityPLOS ONE

Dear Dr. Luppi,

Thank you for submitting your manuscript to PLOS ONE. After careful consideration, we feel that it has merit but does not fully meet PLOS ONE’s publication criteria as it currently stands. Therefore, we invite you to submit a revised version of the manuscript that addresses the points raised during the review process.

The evaluation by the reviewers was mixed. One reviewer underlined the lack of economics analysis in motivating the findings and questioned the role of gender in what seems to be a family decision more than an individual one; the send reviewer is more positive and asked for clarification. My view is somehow in between: I liked the paper and I backing the research project goals but I also think that the economic mechanism behind the outcomes of the data analysis requires a deeper inspection of individual vs household planning and a potential revision of the analysis. I will ask you to take all the suggestions by the referees into account and reply to every point raised.

We look forward to receiving your revised manuscript.

Kind regards,

Luca De Benedictis, PhD

Academic Editor

PLOS ONE

Journal Requirements:

Reviewers' comments:

Reviewer's Responses to Questions

**Comments to the Author**

1. Is the manuscript technically sound, and do the data support the conclusions?

Reviewer #1: Partly

Reviewer #2: Yes

2. Has the statistical analysis been performed appropriately and rigorously? 

Reviewer #1: Yes

Reviewer #2: Yes

3. Have the authors made all data underlying the findings in their manuscript fully available?

Reviewer #1: No

Reviewer #2: No

4. Is the manuscript presented in an intelligible fashion and written in standard English?

Reviewer #1: Yes

Reviewer #2: Yes

5. Review Comments to the Author

Reviewer #1: The manuscript analyses the effects of occupational status and perceived income vulnerability on fertility plans before and during the Covid19 pandemics.

The paper is well structured, clear and easy to read. Anyway, a proofreading is recommended as the text shows a few typos and errors (e.g. p.6 line 132 ‘future worse family income’; line 145 ‘the crisis is negatively influences’; etc). The text in the appendix needs to be checked as well.

Authors find that those affected by a negative shock in their expectations on income and occupation following the pandemic outbreak are more likely to abandon their pre-pandemic fertility plan, with little difference by gender.

However, the contribution and novelty of the outcome are unclear. On the one hand, one of the authors’ main findings (the expected negative impact on future income negatively affects fertility plans) is a straightforward result. On the other hand, analysing fertility plans in 2020 may be poorly relevant now given that actual fertility rates in Italy are available with a more recent date.

Authors claim that respondents to the survey are chosen in such a way to ensure representativeness with respect to a list of variables. However, the samples on which the analysis is built are severely subset. How is representativeness guaranteed in the final subsamples used?

In addition, I have serious doubts that make me question the meaning of the analysis. This study appears to me as a mere statistical exercise that lacks socio-economic intuition and interpretation.

I appreciate that the survey on which this study relies on is not self-drafted -or is it? Given that at least some of the authors seem to be involved in drafting the Rapporto Giovani this is not clear-. In any case, I believe that the question on which the ‘dependent variable based on the 2016 and 2020 Regular Waves’ relies is poorly designed in its answers. I like the ‘forced’ 4-point Likert scale to avoid neutral answers, but “maybe not” and “maybe yes” seem pretty much the same answer to me. These answers are reported with a different wording (Probably not, Probably yes) in Table 1. Which of the two is correct? I believe the two different wordings bear a significantly different meaning that is not possible to clarify since authors did not attach the original survey to their submission.

In addition, although the survey is submitted to individuals, I find that the question ‘Do you expect to conceive a child within the next 12 months?’ would be more appropriate to be asked to households. If this is the literal translation from the original survey, which I assume to be in Italian, regardless of the respondent’s gender, I think that answers to this question carry the effect of e.g. desires, occupation, job contract’s stability, age of partner. In the way the question is posed it seems much more of a couple's question whose answer is probably the result of common plans, rather than individual ambitions and expectations, especially in a country like Italy where it is rather difficult if not impossible to become a single parent in the short term. Therefore, I would expect controls related to e.g. length of relationship or partner’s variables. As the analysis is currently built, I wonder if it makes sense at all to split it by gender. The fact that there is little difference by gender in the way economic uncertainty seems to have affected fertility plans could be in line with my reasoning. Again, relying on the fact that the question above mentioned is targeted on conception expectations in the short term, I also wonder what sense this makes when asked to individuals with ‘single’ marital status, who represent the majority or about half of the observation samples.

The study completely lacks information about the socio-economic condition of respondents. Do authors have information about actual income? It is not clear since data are not shared. Do the employed respondents work in the private or public sector? A permanent job contract in the private or public sector may make a difference in Italy, but there is no information about this in the paper and again since the survey is not attached there is no way to retrieve this bit of information. Also, I remain sceptical about clustering together employees with permanent and non-permanent contracts. This may have meaning in countries other than Italy, but it is questionable in the Italian context. The study ignores relevant features of the Italian labour market.

Excluding full-time students is not enough to ensure a homogeneous sample. For instance, I believe that a 22-year-old with no tertiary education cannot be considered homogeneous to a 30-year-old new to the job market who completed a long educational path. Restricting the respondents’ age to 18-34 is another limitation. Authors do not consider that fertility plan of educated individuals may start to develop at a later age in Italy.

Before this manuscript could be considered for publication, I invite the authors to reflect on my comments, re-design their analysis consequently, and make sure to attach an accurate translation of the questionnaire or at least report the link or repository where this is available, in case it is accessible.

Reviewer #2: Review of the manuscript titled ”Fertility plans during the COVID-19 pandemic in Italy: the role of occupation and income vulnerability”. The study investigates the role of economic uncertainty for the change in fertility intentions due to the Covid-19 pandemic in Italy. It utilizes two sources of data: two cross-sectional surveys collected in 2016 and November 2020, as well another two cross-sectional surveys collected in March and October 2020, and logistic regression as the method. Their main findings show that fertility intentions have declined among those in an occupational situation characterized by uncertainty, but not among others, and that intentions were abandoned more often in March than October. The study provides some new insights on the impacts on covid-19 on fertility, but also has shortcomings and could benefit from further development.

Main comments:

The authors should better provide the rationale for their research design. Firstly, it should be better articulated what the authors aim at capturing with the two sets of surveys, each with two waves? For instance, to what extent should the change between 2016 and 2020 be interpreted as an effect of the pandemic? Or is this analysis meant to set the stage for the other analysis? There is quite a long time period between the 2016 and 2020 surveys, and it is clear that other factors than only the pandemic can intervene. Second, how should the comparison between the two surveys during the pandemic be interpreted? For instance, if both make a reference to the fertility plans before the pandemic started, then would it be natural to expect that there is more change from the original plan as time passes?

The categorization of the occupational grouping seems particular (see l. 298-319), as most of the employed are categorized in one group. I think this is fine, but the particular nature of the categorization should be highlighted. It is surprising that those with a fixed-term and permanent contract are grouped together, and I feel showing results while breaking this group into two in the Appendix may be necessary, as it provides insights as well. NEET and self-employed are particular groups among the employed. It is unclear why professionals and managers are particularly mentioned as part of the third group. Lines 313-319 mention about this choice being consistent with some results from year 2020 but is not clear with which results exactly. Moreover, at least in October-November, the pandemic may already have affected the occupational condition of the respondent; please discuss. The self-reported information on changes in economic situation then seems valuable also because of this. Also, in my view the distribution of the main explanatory variables should be shown in the main body of the article.

Related to the earlier comment, there could be more consistency in the terminology used, e.g. occupational and income vulnerability, employment condition, occupational condition, “experienced and expected income and occupational vulnerabilities”. In the same vein, the title of the study could use some editing (e.g. “occupation” -> “occupational”).

Why is uncertainty so differently related to postponing and abandoning? This would require some discussion to validate and understand the current results. For instance, l. 463 -464: “Postponers do not show any significant path, with a slightly – even though not significant – similar behaviour to that of the still-planners.” See also lines 481-482

Other comments:

The expectations regarding gender differences should be introduced in the front part of the paper.

l.62-65: At which level of analysis is the argument relevant, please specify. For instance, incidence may be correlated with the severity of the economic shock at the country level.

l.71-72: provide a reference for unequal effects of the pandemic

l. 179-181: The authors overlook heterogeneity in responses to the pandemic, as shown for instance by Aasve et al. 2021., not all though most of the evidence points to a negative effect of the pandemic on fertility. Perhaps more importantly, the authors should here report previous findings of the impact of the pandemic on fertility intentions in high-income countries, rather than going directly to evidence from the low- and middle-income countries.

l. 199-201: It is not entirely clear what heterogeneity here refers to, perhaps not the best term here.

The response rates of the different surveys are currently not reported. They should be, as the selective response may affect the current findings especially when comparing probabilities across surveys.

The description of the dependent variable could be clearer, e.g. (l.280-282) “In March, this question has been asked to both those who – in January 2020 – declared strong [3a] and weaker intentions [2a] to conceive a child during the next 12 months” can be understood as if there was a survey in January too.

l. 283-286: It is not clear why the sample is restricted based on different criteria (weaker and stronger intentions) in the two surveys during the pandemic, please justify better.

l. 286-289: It is not clear what is meant here with “selection”, and in which way the robustness analysis is supposed to take care of this, and the additional results are not interpreted in the text. Please clarify.

l. 384: Please use the term “short-term” more consistently. Perhaps it could refer to the next 12 months only?

l. 420-442: “Because we do not find significant variations for the other categories of the outcome variable, Figure 1 shows the predicted probabilities corresponding to the extreme category “surely yes”.” Please elaborate on this choice. The ordered logit estimates a coefficient for the change from one category to the next.

l. 590-593: In which way is it beneficial for policy makers to know about fertility intentions in times of great uncertainty? What capacities do they have for reducing such uncertainty? I’d suggest being more cautious

Figure 2-5: The titles require editing. Specify also that the following refers to: “with confidence intervals for pair-wise comparisons at the 10% significance level”

Figures 3-5: It seems the panel labels are not correctly placed or are missing. This might also help the reader in the interpretations of the results, which seem at times a bit difficult as based on the text. The author could also specify here which survey wave is used.

Tables: No need to include “%“ after each number in the table.

6. PLOS authors have the option to publish the peer review history of their article (what does this mean?). If published, this will include your full peer review and any attached files.

Reviewer #1: No

Reviewer #2: No

---

## [Author Response · Author response to Decision Letter 0]

5 Jun 2022

Dear prof. Luca de Benedectis, Editor of PLOSONE, 

Dear Reviewers,

we are grateful for the thoughtful and useful comments on our manuscript. In response to this helpful feedback, we have carefully revised the paper and implemented new analyses. We believe that Reviewers’ suggestions have been essential for improving the quality of the paper. 

In what follows, we summarize the changes made in the manuscript and we report our point-to-point replies to the Editor’s and Reviewers’ comments.

Looking forward to receiving your decisions about our manuscript

Our best regards,

Francesca Luppi

Bruno Arpino

Alessandro Rosina

Response to the Reviewers

Reviewer #1: The manuscript analyses the effects of occupational status and perceived income vulnerability on fertility plans before and during the Covid19 pandemics.

The paper is well structured, clear and easy to read. Anyway, a proofreading is recommended as the text shows a few typos and errors (e.g. p.6 line 132 ‘future worse family income’; line 145 ‘the crisis is negatively influences’; etc). The text in the appendix needs to be checked as well.

Authors find that those affected by a negative shock in their expectations on income and occupation following the pandemic outbreak are more likely to abandon their pre-pandemic fertility plan, with little difference by gender.

ANSWER. We proofread the text.

POINT 1. However, the contribution and novelty of the outcome are unclear. On the one hand, one of the authors’ main findings (the expected negative impact on future income negatively affects fertility plans) is a straightforward result. On the other hand, analysing fertility plans in 2020 may be poorly relevant now given that actual fertility rates in Italy are available with a more recent date.

ANSWER. Following the Reviewer’s suggestion, now we stress more the contribution of our research in the Introduction and in the Discussion sections. 

First, there are still very few studies based on micro-level data analysing the possible impact of the COVID-19 crisis on fertility, and almost all of them deals with fertility intentions (e.g., Zimmerman, L. A., Karp, C., Thiongo, M., Gichangi, P., Guiella, G., Gemmill, A., ... & Bell, S. O. (2022). Stability and change in fertility intentions in response to the COVID-19 pandemic in Kenya. PLOS Global Public Health, 2(3); Emery, T., & Koops, J. C. (2022). The impact of COVID-19 on fertility behaviour and intentions in a middle income country. Plos one, 17(1), e0261509; Sienicka, A., Pisula, A., Pawlik, K. K., Kacperczyk-Bartnik, J., Bartnik, P., Dobrowolska-Redo, A., & Romejko-Wolniewicz, E. (2021). The impact of COVID-19 pandemic on reproductive intentions among the Polish population. Ginekologia polska.; Malicka, I., Mynarska, M., & Świderska, J. (2021); Perceived consequences of the COVID-19 pandemic and childbearing intentions in Poland. Journal of Family Research, 33(3), 674-702). At the same time, while there is some evidence based on birth rates about an impact of the COVID-19 crisis on fertility at the country level, macro level data do not allow to explore the relationship with (for example) perceived and experienced occupational and income vulnerability. Our results, in this sense, help to shed light on the macro-level evidence of a birth decline in Italy during the pandemic as reported in other studies. We now report some references to macro-level studies on birth rates in both the Introduction and Background sections.

Moreover, as we stress in the Introduction, fertility intentions and expectations are strong predictors of fertility behaviours (e.g., Billari et al. 2008; Régnier-Loilier and Vignoli 2011; Schoen et al. 1999), while offering the great advantage to promptly provide evidence of a possible impact of the crisis on fertility plans. This literature also stresses the importance of exploring fertility intentions to better understand the decision-making process behind fertility outcomes. The way fertility intentions are operationalized in our study is even more informative in this sense. In fact, next to a traditional formulation, i.e., the intention to conceive within 12 months, we use a second measure, which is the intention to confirm, postpone or abandon pre-COVID fertility plans. In this way, we “offer insights on possible mechanisms leading to fertility postponement for which fertility behaviours would provide evidence only several years after the end of the pandemic”. In other words, this kind of fertility intentions offers a retrospective information which is usually missing in cross-sectional studies (even when studying fertility behaviours). Also in this case, we can consider how individuals differ in their decision to confirm or change their original fertility plans because of the pandemic, highlighting the features of those more negatively impacted by the crisis (which again is still not possible by using macro-level birth or fertility rates). Thus, our analyses allow examining changes in fertility plans that would be impossible to do with macro-level data on fertility rates. The usefulness of looking at fertility intentions is also confirmed by several recent papers on the effect of the Great recession of 2008 that still use individual data on fertility intentions despite the availability of macro – and micro-level data on fertility behaviours (e.g., Vignoli, D., Mencarini, L., & Alderotti, G. (2020). Is the impact of employment uncertainty on fertility intentions channeled by subjective well-being. Advances in Life Course Research.; Novelli, M., Cazzola, A., Angeli, A., & Pasquini, L. (2021). Fertility Intentions in Times of Rising Economic Uncertainty: Evidence from Italy from a Gender Perspective. Social Indicators Research, 154(1), 257-284.). Even though data on fertility behaviours are largely available for those years, fertility intentions are more informative about the mechanisms behind the decision to conceive, as they represent the very first reaction to – for example as in this case – an economic shock. A discussion on this issue is now reported in the Introduction and Conclusion sections.

Regarding the association between fertility plans and subjective and objective indicators of employment and income vulnerability, we now try to better present the important contribution of our results to the knowledge of the mechanisms linking the COVID-19 crisis to fertility. First, we consider the recent literature on “economic uncertainty” which highlights the need to use both objective and subjective indicators to describe this kind of vulnerability. This is consistent also with the fact that the objective signs of the economic recession (e.g., increasing unemployment rates, income loss, etc.) - both at macro and micro level - arise only later in the 2020. Thus, “the availability of data collected at two time points in the 2020 allows to distinguish between a very early impact of the crisis (in March, when the dread of a possible global recession and the shock of the first lockdown were the possible predominant drivers) to a later one (in October, when the recession signs were already evident, while the summer break had relieved the worries induced by the health emergency)”. In this sense, now we better specify the fact that we are focusing on a specific period of the COVID-19 crisis, which is the early phase (in the title, introduction, and conclusion). 

As previously stated, the use of both objective and subjective indicators of economic uncertainty allows to explore how the economic recession might have differently impacted on fertility plans according to different moments of the COVID-19 crisis and by gender. We basically found that objective and subjective indicators are not associated in the same way with fertility plans, with more vulnerable occupational conditions associated with suspended fertility plans only in March 2020 but not later in the year; a worse financial situation and expected a negative effect on the individual’s occupation is more associated to the decision to suspend the fertility plans among women than among men; while both men and women negatively revised their pre-pandemic fertility plans when they expect a worsening in the financial situation. 

 

POINT 2. Authors claim that respondents to the survey are chosen in such a way to ensure representativeness with respect to a list of variables. However, the samples on which the analysis is built are severely subset. How is representativeness guaranteed in the final subsamples used? 

ANSWER. The Reviewer correctly highlights the need for more clarity on this aspect. First of all, we stress now in the conclusion the limitation implied by the facto of using data from surveys conducted on quota samples, which guarantee the representativeness only with respect to a set of variables on which quota have been defined. Moreover, cases are always weighted in all the descriptive analyses and in the models. 

Only for the Covid survey, because we exclude from the main analysis individuals without pre-pandemic fertility plans, our final sample is selected, and we cannot exclude that our dependent variables (i.e., objective and subjective uncertainty) might guide this selection process (e.g., those who do not declare to have pre-pandemic fertility plans might be already experiencing or expecting financial and occupational uncertainty, independently by the occurrence of the COVID-19 crisis). In the Appendix we provide some models which results basically show that “the probability of being no-planners is higher when people perceive their income at risk, for both genders (Model 3) and for men also in the case in which they already experience an income loss (Model 4). However, when occupation is seen as vulnerable, this affects fertility plans more on the female than on the male side (Model 2)”. Thus, our final sample is a selection of respondents that excludes vulnerable individuals in terms of occupation and financial conditions. 

Finally, by excluding full-time students, we are further restricting our sample. In this case we re-run all the models by re-including full-time students (as a separated occupational category): results are always stable (results from the models with full-time students are now reported in the Appendix, Robustness check section: see Figures from S1_2 to S1_5). Thus, as excluding full-time students is not affecting our results, we can assume that, at least for the variables we are considering, the sample distributions are not significantly modified. A short discussion on that is reported in the paper.

POINT 3. In addition, I have serious doubts that make me question the meaning of the analysis. This study appears to me as a mere statistical exercise that lacks socio-economic intuition and interpretation.

ANSWER. As also discussed in reply to others Reviewer’s comments, we have now revised the introduction to better highlight the importance and contribution of our study. First, we refer to the general literature on fertility intentions and expectations to justify our focus on these aspects rather than on macro-level fertility outcomes. As mentioned at POINT 1, fertility intentions are considered as the most proximate determinants of fertility outcomes and examining them allow to better understand the decision-making process behind fertility outcomes. Second, we state more clearly our contributions to the on-going literature on the consequences of the COVID-19 pandemic on fertility. In this respect, we use unique data on several phases of the pandemic that allow us to assess whether the (additional) economic uncertainty created by the pandemic created only a temporary effect in the very early phases or could be detectable also later. In addition, although not perfect, our comparison with pre-COVID data allow to better appreciate the changes brought by the pandemic. Finally, we use several objective and subjective indicators of economic uncertainty, and we show that their associations with fertility plans change by gender and by different times of the COVID crisis. We now stress more these points along the paper.

 

POINT 4. I appreciate that the survey on which this study relies on is not self-drafted -or is it? Given that at least some of the authors seem to be involved in drafting the Rapporto Giovani this is not clear-. In any case, I believe that the question on which the ‘dependent variable based on the 2016 and 2020 Regular Waves’ relies is poorly designed in its answers. I like the ‘forced’ 4-point Likert scale to avoid neutral answers, but “maybe not” and “maybe yes” seem pretty much the same answer to me. These answers are reported with a different wording (Probably not, Probably yes) in Table 1. Which of the two is correct? I believe the two different wordings bear a significantly different meaning that is not possible to clarify since authors did not attach the original survey to their submission. 

ANSWER. We thank the reviewer to have noticed the inconsistency: the right translation from the Italian is “Probably not” and “Probably yes”. The question wording and the answer options are identical as those in the Italian “Famiglie e Soggetti Sociali” survey, the largest survey conducted in Italy by the national institute of statistics (ISTAT) on household and family issues. These questions are also standard in the international research on fertility intention (see e.g., the Gender and Generation Survey). This has the advantage of allowing possible inter-survey comparisons. Now some extracts from the questionnaires of the regular and COVID surveys (Italian and English version) and the minimal dataset are available.

POINT 5. In addition, although the survey is submitted to individuals, I find that the question ‘Do you expect to conceive a child within the next 12 months?’ would be more appropriate to be asked to households. If this is the literal translation from the original survey, which I assume to be in Italian, regardless of the respondent’s gender, I think that answers to this question carry the effect of e.g., desires, occupation, job contract’s stability, age of partner. In the way the question is posed it seems much more of a couple's question whose answer is probably the result of common plans, rather than individual ambitions and expectations, especially in a country like Italy where it is rather difficult if not impossible to become a single parent in the short term. Therefore, I would expect controls related to e.g., length of relationship or partner’s variables. 

ANSWER. We agree with the Reviewer that adding the information on the partners’ characteristics would a more thorough investigation. Unfortunately, we do not have this kind of information. Only for the 2016 and 2020 regular surveys we can control for the fact that the individual is in a stable relationship (not necessarily married) and for the length of this relationship. However, by selecting only those in a stable relationship and controlling for its duration results do not change from the ones shown in Figure R.1 (see figure below). See answer to POINT 6 for a further discussion on this issue.

 

Figure R.1. Predicted probabilities of intending to have a(nother) child (answer: “surely yes”) in the next 12 months by occupational condition in 2016 and 2020 (pooled waves) and by gender (confidence intervals for pair-wise comparisons at the 10% significance level)

POINT 6. As the analysis is currently built, I wonder if it makes sense at all to split it by gender. The fact that there is little difference by gender in the way economic uncertainty seems to have affected fertility plans could be in line with my reasoning. Again, relying on the fact that the question above mentioned is targeted on conception expectations in the short term, I also wonder what sense this makes when asked to individuals with ‘single’ marital status, who represent the majority or about half of the observation samples. 

ANSWER. The Reviewer is right when highlighting the implications of considering fertility expectations instead of intentions. According to the Miller and Pasta’s Traits-desire-intentions-behaviours model (TDIB), expectations are in-between intentions and behaviours (e.g., Miller, W., Severy, L., and Pasta, D.J. (2004). A framework for modelling fertility motivation in couples. Population Studies 58(2): 193–205). Asking about fertility expectations, instead of intentions, has the advantage of emphasizing the “likelihood” of realizing fertility plans (Warshaw and Davis 1985): short-term expectations take into account situational factors that can impact on the chances of realizing someone’s intention to conceive (Heiland, Prskawetz and Sanderson 2008), including the partner’s situation, intentions and preferences. Nonetheless, expectations’ determinants tend to be similar to those of intentions (Gray, Evans, and Reimonds 2013) and a clearer distinction between the two concepts is often absent in the demographic literature (see Luppi and Mencarini 2019 for a discussion on that). For this reason, in a cautious way, we decided to treat fertility expectations similarly to intentions, and so keeping separated women and men. 

About the interpretation of the gendered patterns in our results, only in few cases differences between genders are significant (at least for the COVID surveys). We might consider this as a possible confirmation of the assumption that fertility expectations (and plans) take into account also the partners’ features; or that there are no gender differences in the way occupational (and income) vulnerability affects fertility expectations (and plans) among individuals below their 35s. As we cannot test which of the two interpretations is correct, we believe the by gender approach is necessary. As suggested by Reviewer 2, we add in the Introduction some possible expectations about gendered patterns in our results.

About the issue of “single” individuals, we never refer to the “other” category of the variable “married” as single individuals. However, we understand the Reviewer’s doubt, as at one point of the manuscript we erroneously reported the variable operationalization, by including cohabiting individuals in the category of “married” ones: actually, what we do is simply distinguishing between married and not married individuals, where “not married” can be either cohabiting couples, LAT (living apart together), or unpartnered individuals. We have reasons to think that single-individuals do not represent the majority of our samples, neither of the “others” category. In Italy, in particular, we expect to have a high proportion of LAT, i.e., young people in partnership still living with their parents until they marry. Because being married is still often a precondition for conceiving in Italy (see Castagnaro and Prati (2022) in L’IMPATTO DELLA PANDEMIA DI COVID-19 SU NATALITÀ E CONDIZIONE DELLE NUOVE GENERAZIONI Secondo rapporto del Gruppo di esperti “Demografia e Covid-19”, available at: https://famiglia.governo.it/media/2671/secondo-report_gde-demografia-e-covid-19_finale.pdf ), the decline in the marriage rate in the 2020 (because of the pandemic) might have enlarged the relative number of individuals with pre-pandemic intentions to conceive among those who are not married yet at the survey time (but they may be in a stable relationship tough). However, because of the type of dependent variables we are using a (i.e., short-term fertility expectations in the regular surveys, and revised pre-pandemic plans of conceive a child in the 2020 in the COVID surveys), we mostly selected partnered individuals, excluding the majority of single-persons (which hardly answer they intention to conceive in three years, a precondition for answering the question about 12 months fertility expectations). An empirical confirmation of that is reported in the answer to POINT 5 for the 2016/2020 data, where we selected only partnered persons and we found that results did not change. In absolute number, only 54 individuals in the 2016 and 35 in the 2020 who are not in a stable relationship declare they surely intend to have a child in the next 12 months. About the COVID survey, we do not have information about the fact of being in a stable relationship, a part of being married (thus, we cannot distinguish single persons from not-married partners). Nevertheless, because of what discussed above, we feel safe in interpreting the impact of single-persons’ plans as negligible. 

POINT 7. The study completely lacks information about the socio-economic condition of respondents. Do authors have information about actual income? It is not clear since data are not shared. Do the employed respondents work in the private or public sector? A permanent job contract in the private or public sector may make a difference in Italy, but there is no information about this in the paper and again since the survey is not attached there is no way to retrieve this bit of information. 

ANSWER. Unfortunately, we do not have such information in the survey. We now discuss this limitation in the conclusion section.

POINT 8. Also, I remain sceptical about clustering together employees with permanent and non-permanent contracts. This may have meaning in countries other than Italy, but it is questionable in the Italian context. The study ignores relevant features of the Italian labour market. 

ANSWER. We thank the Reviewer for rising the point. Now, we better justify our choice of clustering together, both on interpretative and empirical terms. 

This clustering strategy is driven by analytical reasons: the small sample size requires parsimonious categorization of the independent variables. However, our choice is supported not only by the empirical evidence, but also by some interpretative reasons.

First, we have improved the interpretation of the “occupational status” variable as “occupational vulnerability” to the economic recession, in terms of both earning and employment. The NEET category is the most vulnerable in this sense, as people in this cluster are probably experiencing the higher difficulties (and lower probability) in gathering an employment in periods of economic crisis and they do not have an own earning. Those working as self-employed or precarious workers experience lower levels of occupational vulnerability compared with NEET but higher if compared with the residual category: while they work and earn, they also experience a high risk because of their more vulnerable income/employment protection during the pandemic period. Moreover, compared to employees with non-permanent contracts, they tend to have lower social security contributions and they do not benefit from employment protection legislation. Finally, the non-permanent and permanent employees are clustered together with the professional and manager as they report higher levels of earnings, higher occupational stability and/or higher levels of employment protection. This is particularly valid for the 2020 because of layoffs ban measure as now discussed also in the Introduction. 

Our clustering choice are supported also by empirical results. Next to the results of the tests already reported in the paper (“i.e., tests of differences between proportions did not reject the null hypothesis of equality between the proportions of still-planning/postponing/abandoning individuals between the two groups of employees with and without a permanent contract”), we add in the S1_Appendix the same model as reported in Figure 2, by keeping all the occupational categories as much disaggregated as possible (see Figure S1_1 in the S1_Appendix). We pool men and women together, as well as the two waves, in order to have a sufficient number of cases within each occupational category. The predicted probabilities, especially those for the “abandoning” and “still-planning” outcomes, support our clustering decisions, by showing that fertility plans are quite homogeneous among temporary and permanent employees and professional/managers. On the same line, also self-employed and free-lancers (i.e., precarious workers) are rather similar. 

Even though we agree on the fact that these working categories differ for many aspects, they seem to behave similarly with respect to fertility plans at least during the early phase of the pandemic. We interpret this result as a confirmation of the fact that our variable is not mirroring the occupational status but the (experienced or perceived) income/employment vulnerability associated with that. 

POINT 9. Excluding full-time students is not enough to ensure a homogeneous sample. For instance, I believe that a 22-year-old with no tertiary education cannot be considered homogeneous to a 30-year-old new to the job market who completed a long educational path. Restricting the respondents’ age to 18-34 is another limitation. Authors do not consider that fertility plan of educated individuals may start to develop at a later age in Italy. 

ANSWER. The Reviewer is right in stressing the limit of focusing only on young individuals. We already highlighted this limitation in our conclusion, stating that, however, young individuals in Italy seem to be the ones most affected by the economic recession in terms of reduced employment opportunities and income. The mean age at birth in Italy is around 32, which means that we include in our sample the typical ages at which childbearing plans are realized. For sure there is a selection in that, as those with higher levels of education report higher age at (first) birth: unfortunately, we cannot address this issue other than including the standard control variables for that. 

However, we agree that excluding full-time students does not make our sample more homogeneous (we have removed this statement now). In the S1_Appendix we report now the same analyses done on the sample including also the full-time students’ category (see Figures S1_2 to S1_5). Results are always stable (i.e., consistent with the results from the samples without full-time students, as reported in the main manuscript). 

POINT 10. Before this manuscript could be considered for publication, I invite the authors to reflect on my comments, re-design their analysis consequently, and make sure to attach an accurate translation of the questionnaire or at least report the link or repository where this is available, in case it is accessible. 

ANSWER. The full questionnaires are not available, but, together with a minimal dataset, we add some extracts from them, including the questions from which the variables included in the analyses are derived. Both the Italian original version of the questions and their English translation are provided. 

Reviewer #2: Review of the manuscript titled ”Fertility plans during the COVID-19 pandemic in Italy: the role of occupation and income vulnerability”. The study investigates the role of economic uncertainty for the change in fertility intentions due to the Covid-19 pandemic in Italy. It utilizes two sources of data: two cross-sectional surveys collected in 2016 and November 2020, as well another two cross-sectional surveys collected in March and October 2020, and logistic regression as the method. Their main findings show that fertility intentions have declined among those in an occupational situation characterized by uncertainty, but not among others, and that intentions were abandoned more often in March than October. The study provides some new insights on the impacts on covid-19 on fertility, but also has shortcomings and could benefit from further development.

Main comments:

POINT 1. The authors should better provide the rationale for their research design. Firstly, it should be better articulated what the authors aim at capturing with the two sets of surveys, each with two waves? For instance, to what extent should the change between 2016 and 2020 be interpreted as an effect of the pandemic? Or is this analysis meant to set the stage for the other analysis? There is quite a long time period between the 2016 and 2020 surveys, and it is clear that other factors than only the pandemic can intervene. Second, how should the comparison between the two surveys during the pandemic be interpreted? For instance, if both make a reference to the fertility plans before the pandemic started, then would it be natural to expect that there is more change from the original plan as time passes?

ANSWER. The Reviewer is right in questioning whether we can attribute the variation in fertility intentions between 2016 and 2020 to a “pandemic effect”. Actually, we cannot. The aim of this first part of the analysis is to explore whether the results obtained by using the traditional operationalization of the fertility intentions mirrors the ones arisen by using the “revised fertility intentions” formulation. This will offer a link to other studies carried on using the traditional fertility intentions formulation. In this sense, this step of the analysis is introductory to the “core”.

About the issue regarding the comparison between the COVID surveys, the availability of two waves of the same survey, conducted at different moment of the health emergency, allows to enlarging the sample size and test whether the pandemic effect differs from the very beginning of the crisis (when the global recession was only forecasted) to some months later, after the summer break and right before the beginning of the second pandemic wave (when the recession was already in action). 

Actually, according to our descriptive results in Table 2, the relative number of those postponing/abandoning the plans declines in October 2020 and it is lower when compared to March of the same year. Meaning that, probably, respondents were more optimistic in October than in March. Additionally, our main interest is in the role of experienced or expected economic uncertainty in changing or confirming of pre-covid fertility plans, rather than the changes in the incidence of those modifying their original intentions.

All these points have been better clarified both in the Introduction and in the Discussion sections.

POINT 2. The categorization of the occupational grouping seems particular (see l. 298-319), as most of the employed are categorized in one group. I think this is fine, but the particular nature of the categorization should be highlighted. It is surprising that those with a fixed-term and permanent contract are grouped together, and I feel showing results while breaking this group into two in the Appendix may be necessary, as it provides insights as well. NEET and self-employed are particular groups among the employed. It is unclear why professionals and managers are particularly mentioned as part of the third group. Lines 313-319 mention about this choice being consistent with some results from year 2020 but is not clear with which results exactly. 

Moreover, at least in October-November, the pandemic may already have affected the occupational condition of the respondent; please discuss. The self-reported information on changes in economic situation then seems valuable also because of this.

Also, in my view the distribution of the main explanatory variables should be shown in the main body of the article. 

ANSWER. We thank the Reviewer for rising the point. Now, we better justify our choice of clustering together, both on interpretative and empirical terms. 

This clustering strategy is driven by analytical reasons: the small sample size requires parsimonious categorization of the independent variables. However, our choice is supported not only by empirical evidence, but also by some interpretative reasons.

First, we have improved the interpretation of the “occupational status” variable as “occupational vulnerability” to the economic recession, in terms of both earning and employment. The NEET category is the most vulnerable in this sense, as people in this cluster are probably experiencing the higher difficulties (and lower probability) in gathering an employment in periods of economic crisis and they do not have an own earning. Those working as self-employed or precarious workers experience lower levels of occupational vulnerability compared with NEET but higher if compared with the residual category: while they work and earn, they also experience a high entrepreneur risk because of their more vulnerable income/employment protection during the pandemic period. Moreover, compared to employees with non-permanent contracts, they tend to have lower social security contributions and they do not benefit from employment protection legislation. Finally, the non-permanent and permanent employees are clustered together with the professional and manager as they report higher levels of earnings, higher occupational stability and/or higher levels of employment protection. This is particularly valid for the 2020 because of layoffs ban measure, as now discussed also in the Introduction. 

Our clustering choice are supported also by empirical results. Next to the results of the tests already reported in the paper (“i.e., tests of differences between proportions did not reject the null hypothesis of equality between the proportions of still-planning/postponing/abandoning individuals between the two groups of employees with and without a permanent contract”), we add in the Appendix the same model as reported in Figure 2, by keeping all the occupational categories as much disaggregated as possible (see Figure S1_1 in the S1_Appendix). We pool men and women together, as well as the two waves, in order to have a sufficient number of cases within each occupational category. The predicted probabilities, especially those for the “abandoning” and “still-planning” outcomes, support our clustering decisions, by showing that fertility plans are quite homogeneous among temporary and permanent employees and professional/managers. On the same line, also self-employed and free-lancers (i.e., precarious workers) are rather similar. 

Even though we agree on the fact that these working categories differ for many aspects, they seem to behave similarly with respect to fertility plans at least during the early phase of the pandemic. We interpret this result as a confirmation of the fact that our variable is not mirroring the occupational status but the (experienced or perceived) income/employment vulnerability associated with that, at least during the 2020. 

Then, the Reviewer is right in saying that the occupational status might have been already changed because of the pandemic in October 2020. This is the reason why we account for more indicators for encompassing economic uncertainty. We are more explicit on this point now in the Introduction and in the Conclusion.

About the distribution of the covariates, Table 1A and 2A are now reported in the main text, as Table 3 and 4, in the Descriptive results section. 

POINT 3. Related to the earlier comment, there could be more consistency in the terminology used, e.g., occupational and income vulnerability, employment condition, occupational condition, “experienced and expected income and occupational vulnerabilities”. In the same vein, the title of the study could use some editing (e.g., “occupation” -> “occupational”).

ANSWER. We revised the manuscript to be consistent along the manuscript with the use of the terminology

POINT 4. Why is uncertainty so differently related to postponing and abandoning? This would require some discussion to validate and understand the current results. For instance, l. 463 -464: “Postponers do not show any significant path, with a slightly – even though not significant – similar behaviour to that of the still-planners.” See also lines 481-482

ANSWER. The Reviewer is right in noticing that we did not discuss enough the results for the postponers group. Actually, it seems they are in between the other two groups (i.e., still-planners and abandoners), and sometimes they even share more similarities with those who confirmed the original plan. On the one hand this might be a quite mixed group, with someone postponing for few months and then realizing the plan, while for others the postponement might be longer and a bit more similar to an indefinite suspension of the plan. On the other hand, the fact that postponers seem to be more similar to still-planners than to the abandoners group might be because those who abandoned the plan adopt a very extreme position compared to those that simply postpone but do not quite the program. 

Other comments:

The expectations regarding gender differences should be introduced in the front part of the paper.

ANSWER. We add the following paragraph in the Introduction section. “Finally, gender differences are also explored: in contexts where traditional gender culture is still widespread, economic recessions tend to negatively impact especially on men’s side through an income effect (REF: childbearing represents an additional economic risk, which should be avoided), while it is more common among women the rise of a substitution effect (REF: in times of economic recession, investing in childbearing and care tasks can reduce the overall perceived uncertainty).”

l.62-65: At which level of analysis is the argument relevant, please specify. For instance, incidence may be correlated with the severity of the economic shock at the country level.

ANSWER. We clarify the link, as also suggested by the Reviewer.

l.71-72: provide a reference for unequal effects of the pandemic

ANSWER. References have been added

l. 179-181: The authors overlook heterogeneity in responses to the pandemic, as shown for instance by Aasve et al. 2021., not all though most of the evidence points to a negative effect of the pandemic on fertility. Perhaps more importantly, the authors should here report previous findings of the impact of the pandemic on fertility intentions in high-income countries, rather than going directly to evidence from the low- and middle-income countries.

ANSWER. We changed the introduction and the background section according to the Reviewer’s suggestion. 

l. 199-201: It is not entirely clear what heterogeneity here refers to, perhaps not the best term here.

ANSWER. We removed the term heterogeneity.

The response rates of the different surveys are currently not reported. They should be, as the selective response may affect the current findings especially when comparing probabilities across surveys. 

ANSWER. Because IPSOS have a consolidated panel or respondents, their surveys usually guarantee a response rate which – on average - is in between the 40 - 50%.

The description of the dependent variable could be clearer, e.g. (l.280-282) “In March, this question has been asked to both those who – in January 2020 – declared strong [3a] and weaker intentions [2a] to conceive a child during the next 12 months” can be understood as if there was a survey in January too.

ANSWER. We changed the wording in all the cases implying possible misunderstanding 

l. 283-286: It is not clear why the sample is restricted based on different criteria (weaker and stronger intentions) in the two surveys during the pandemic, please justify better.

ANSWER. We better explained this part. 

l. 286-289: It is not clear what is meant here with “selection”, and in which way the robustness analysis is supposed to take care of this, and the additional results are not interpreted in the text. Please clarify. 

ANSWER. The Reviewer is right: this does not represent a robustness analysis. We simply aim at showing how our sample is selected in terms of economic and employment features, as they probably experience less uncertain conditions then in the entire (representative) sample of young Italians. In other words, we claim that the conditions of economic uncertainty might have play a role even before the start of the pandemic, by selecting the sample of those who were planning to conceive a child before the pandemic. We have tried to better clarify this issue in the paper.

l. 384: Please use the term “short-term” more consistently. Perhaps it could refer to the next 12 months only?

ANSWER. We have carefully revised the text to be consistent.

l. 420-442: “Because we do not find significant variations for the other categories of the outcome variable, Figure 1 shows the predicted probabilities corresponding to the extreme category “surely yes”.” Please elaborate on this choice. The ordered logit estimates a coefficient for the change from one category to the next. 

ANSWER. We estimated ordered logit models but instead of representing coefficients (on the log-odds scale) which, as correctly noticed by the Reviewer, compare one category with the previous, we estimated predicted probabilities each outcome’s category by the values of the explanatory variables. However, instead of reporting all predicted probabilities, we only show those for which a statistically significant difference across the groups defined by the explanatory variables is found. We are sorry we were not clear in the text. We revised it to clarify this aspect.

l. 590-593: In which way is it beneficial for policy makers to know about fertility intentions in times of great uncertainty? What capacities do they have for reducing such uncertainty? I’d suggest being more cautious 

ANSWER. We agree with the Reviewer: we removed this sentence and modified a bit our conclusion.

Figure 2-5: The titles require editing. Specify also that the following refers to: “with confidence intervals for pair-wise comparisons at the 10% significance level”

ANSWER. We are clearer now on this point, and we edit the figures’ titles and notes.

Figures 3-5: It seems the panel labels are not correctly placed or are missing. This might also help the reader in the interpretations of the results, which seem at times a bit difficult as based on the text. The author could also specify here which survey wave is used.

ANSWER. The panel labels are not included because the two waves are pooled here, as explained at the end of the “Method” section: “We also tested interactions between the main covariates and the wave indicator but did not find statistically significant results except for the occupational status. Therefore, while for the subjective indicators of economic uncertainty results are presented without including the interaction with the wave, for the occupational status the results for the interaction terms are shown.”. In order to improve the understanding of the figures, we now recall this issue in the figures title and when presenting the results.

Tables: No need to include “%“ after each number in the table.

ANSWER. “%” have been removed

---

## [Editor Report · Decision Letter 1]

30 Jun 2022

Fertility plans in the early times of the COVID-19 pandemic: the role of occupational and financial uncertainty in Italy

PONE-D-22-07293R1

Dear Dr. Luppi,

I am pleased to inform you that your manuscript has been judged scientifically suitable for publication and will be formally accepted for publication once it meets all outstanding technical requirements.

Kind regards,

Luca De Benedictis, PhD

Academic Editor

PLOS ONE

---

## [Editor Report · Acceptance letter]

13 Jul 2022

PONE-D-22-07293R1 

Fertility plans in the early times of the COVID-19 pandemic: the role of occupational and financial uncertainty in Italy 

Dear Dr. Luppi:

I'm pleased to inform you that your manuscript has been deemed suitable for publication in PLOS ONE. Congratulations! Your manuscript is now with our production department. 

Kind regards, 

on behalf of

Dr. Luca De Benedictis 

Academic Editor

PLOS ONE